



# Identifying Atmospheric Rivers and their Poleward Latent Heat Transport with Generalizable Neural Networks: ARCNNv1

Ankur Mahesh[1,2], Travis A. O'Brien[3,1], Burlen Loring[4], Abdelrahman Elbashandy[4], William Boos[1,2], and William D. Collins[1,2]

[1]Climate and Ecosystem Sciences Division, Lawrence Berkeley National Laboratory, Berkeley, CA, USA
[2]Department of Earth and Planetary Science, University of California, Berkeley, Berkeley, CA, USA
[3]Department of Earth and Atmospheric Sciences, Indiana University, Bloomington, IN, USA
[4]Computational Research Division, Lawrence Berkeley National Laboratory, Berkeley, CA, USA

**Correspondence:** Ankur Mahesh (ankur.mahesh@berkeley.edu)

**Abstract.** Atmospheric rivers (ARs) are extreme weather events that can alleviate drought or cause billions of dollars in flood damage. By transporting significant amounts of latent energy towards the poles, they are crucial to maintaining the climate system's energy balance. Since there is no first-principles definition of an AR grounded in geophysical fluid mechanics, AR identification is currently performed by a multitude of expert-defined, threshold-based algorithms. The variety of AR detection algorithms has introduced uncertainty into the study of ARs, and the algorithms' thresholds may not generalize to new climate datasets and resolutions. We train convolutional neural networks (CNNs) to detect ARs while representing this uncertainty; we name these models ARCNNs. To detect ARs without requiring new labeled data and labor-intensive AR detection campaigns, we present a semi-supervised learning framework based on image style transfer. This framework generalizes ARCNNs across climate datasets and input fields. Using idealized and realistic numerical models, together with observations, we assess the performance of the ARCNNs. We test the ARCNNs in an idealized simulation of a shallow water fluid, in which nearly all the tracer transport can be attributed to AR-like filamentary structures. In reanalysis and a high-resolution climate model, we use ARCNNs to calculate the contribution of ARs to meridional latent heat transport, and we demonstrate that this quantity varies considerably due to AR detection uncertainty.

## 1 Introduction

Atmospheric rivers (ARs) are extreme weather events that have significant impacts on the climate system and human society. When ARs make landfall, they can be crucial for alleviating drought (Dettinger, 2013). They provide up to fifty percent of US West Coast rainfall and thirty percent of rainfall in Europe (Lavers and Villarini, 2015). In a single year, they caused a billion dollars worth of flood damage (Corringham et al., 2019). The original study on ARs (Zhu and Newell, 1998) found that they account for the vast majority of poleward moisture transport. ARs' latent heat transport can also contribute to extreme heatwaves, such as the record-shattering 2021 Pacific Northwest heatwave (Mo et al., 2022).

In this manuscript, we address two major challenges in AR science. *First, there are a variety of AR detection algorithms, and detection uncertainty must be considered when studying ARs* (O'Brien et al., 2020). While ARs are qualitatively defined





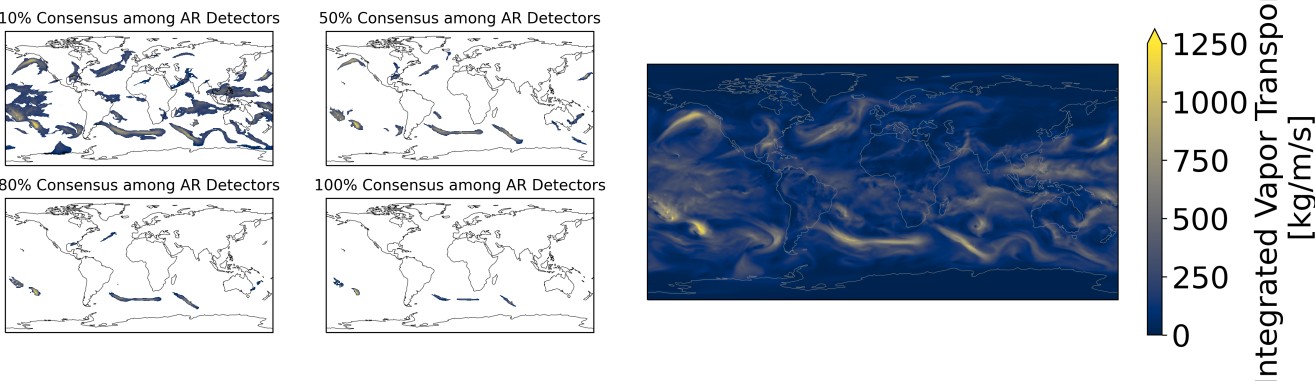

**Figure 1. Variations in AR detections based on level of consensus between AR detectors.** AR detector consensus refers to the proportion of 14 AR detection algorithms that identify an AR at each grid cell. **(left)** Four different levels of consensus on April 4, 2009: 10% (top left), 50% (top right), 80% (bottom left), and 100% (bottom right). Shading represents Integrated Vapor Transport (IVT). **(right)** Global IVT from MERRA2 (Gelaro et al., 2017) on same date.

as long, narrow columns of moisture, there is no unambiguous definition of an AR grounded in geophysical fluid dynamics. As a result, a variety of AR detection algorithms have been developed. In Figure 1, we demonstrate how global AR area varies

significantly as a function of consensus between algorithms. Consensus refers to the proportion of AR detection algorithms that identify an AR at a given grid cell. With a permissive measure of consensus (requiring only 10 percent of detection algorithms to identify an AR), ARs cover a significantly larger fraction of the globe, compared to a more restrictive threshold, such as 80 percent. These differences yield contrasting pictures of global AR activity and AR-induced precipitation, and they pose major implications for a phenomenon-focused understanding of extreme weather. Existing work (Figure 3 of Lora et al. (2020) and

Figure 2 of Rutz et al. (2019)) shows how individual algorithms have different identifications of an AR's spatial extent. The algorithm spread introduces uncertainty in AR size (Inda-Díaz et al., 2021), lifecycle (Zhou et al., 2021), response to climate change (O'Brien et al., 2022), and relationship to the El Niño/ Southern Oscillation (O'Brien et al., 2020). This detection uncertainty is also present in studying other atmospheric phenomena that lack objective definitions, such as blocking (Pinheiro et al., 2019), tropical cyclones (Bourdin et al., 2022), and extratropical cyclones (Neu et al., 2013).

Second, *many AR detection algorithms use criteria that do not generalize across datasets.* In Figure 2, we show detections from the algorithm used in the *Atmospheric Rivers* reference textbook (Ralph et al. (2020), Section 2.3.1). ARs are defined as events poleward of 20 degrees with 250 kg/m/s of Integrated Vapor Transport (IVT), 20 mm of Integrated Water Vapor (IWV), and length scales of at least 2000 km. In Figure 2a, we show that the algorithm identifies an AR in the MERRA2 reanalysis dataset, but it misses a similar AR in a free-running high-resolution climate simulation (Figure 2b). The AR in Figure 2b

lies slightly under the thresholds in IWV and IVT, but it still meets the qualitative definition of an AR. (See Section A in the Appendix for a detailed discussion about why this textbook algorithm does not detect an AR in Fig 2b.) This challenge has been well-identified in the existing literature. Hagos et al. (2015) show that an algorithm's detections vary on different





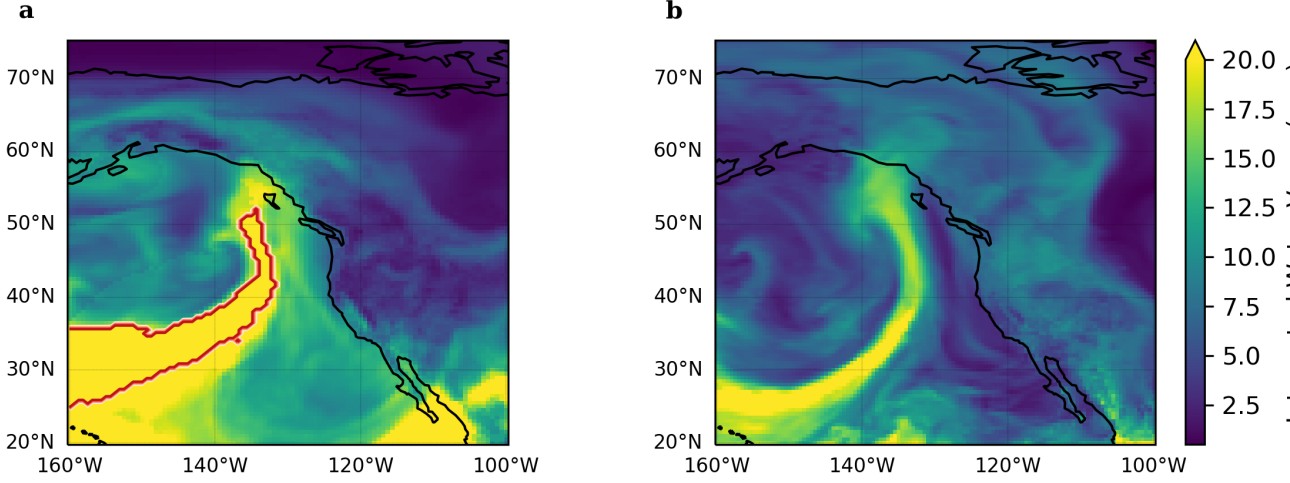

**Figure 2. AR detections in MERRA2 and ECMWF-IFS-HR from a threshold-based detection algorithm. (a)** The detection algorithm from the *Atmospheric Rivers* textbook (Ralph et al., 2020) identifies an atmospheric river (AR) in MERRA2 reanalysis (Gelaro et al., 2017) The AR contour is shown in red. **(b)** The same threshold algorithm does not identify an atmospheric river with a similar structure in ECMWF-IFS-HR (Roberts et al., 2018), a free-running high-resolution climate model.

dynamical cores and resolutions, so the algorithm may need to be revised in light of these differences. Reid et al. (2020) further demonstrate how thresholds in IVT and length are sensitive to dataset resolution. Table 2 of Collow et al. (2022) lists detection
algorithms that needed to be retuned when moving across different datasets with different resolutions. As climate datasets progress (e.g. through the global storm-resolving models as in Stevens et al. (2019)), the scalability and generalizability of AR detection algorithms could pose a challenge for AR detection.

To summarize, for a given dataset, AR detections may differ across algorithms, and for a given algorithm, AR detections may differ across datasets. Because of these two challenges, rigorous assessment of ARs is labor-intensive. It requires retun-
ing, re-implementing, and re-running multiple detection algorithms. Even for some algorithms that themselves do not need to be retuned, they may require re-calculating relative thresholds (e.g. the 85th percentile of IVT) at each grid cell in new datasets. The Atmospheric River Tracking Method Intercomparison Project (ARTMIP) demonstrates the tremendous benefit of systematic AR detection in reanalysis and climate models. Each iteration of ARTMIP requires time, effort, and coordination among many different research groups. On a new dataset, it may be impractical for an individual research group to replicate
this intercomparison.

Therefore, using existing data from ARTMIP, we present convolutional neural networks (CNNs) designed to alleviate the above two challenges. For the first challenge, we train CNNs to replicate the mean of multiple AR detection algorithms, not just one algorithm; we name these neural networks ARCNNs. For the second challenge, to generalize an ARCNN to different climate datasets, we apply a method from computer vision called style transfer. By incorporating detection uncertainty and
generalizability, our ARCNNs build upon prior work that has trained neural networks to detect ARs (Mudigonda et al., 2021;





Kurth et al., 2018; Prabhat et al., 2021; Racah et al., 2017; Liu et al., 2016; Mudigonda et al., 2017) and related weather phenomena, such as warm conveyor belts (Quinting et al., 2022) and extratropical cyclones (Kumler-Bonfanti et al., 2020). Prior CNNs for AR detection have been trained on binary AR detections, and they have not been designed with explicit considerations of generalizing to multiple climate datasets. Through style transfer, our framework generalizes ARCNNs to new datasets without requiring additional training labels, which are a major bottleneck in using machine learning in earth system science (Reichstein et al., 2019). In particular, we train an AR detector to detect ARs in satellite data. Removing the need to identify ARs by hand, this could be deployed for meteorological applications, before reanalyses become available.

A challenge with CNNs is that they have millions of tunable parameters, so they can be hard to interpret. Prior research (Mamalakis et al., 2022a; Madakumbura et al., 2021; Mahesh et al., 2019; Davenport and Diffenbaugh, 2021; Toms et al., 2020) has applied interpretability methods from computer vision to CNNs used in climate science. To validate our CNNs' trustworthiness, we use a hierarchy of dynamical models of varying complexity. We validate our CNNs on an idealized simulation, in which almost all the midlatitude vapor transport can be attributed to AR-like structures. We test whether the CNN's detections meet this characteristic of the simulation.

We demonstrate that ARCNNs can be used to explore important AR-related science questions. We apply ARCNN detector to investigate how detector uncertainty affects estimates of AR-induced meridional latent heat transport (LHT). Traditionally, ARs have been thought to account for virtually all the meridional moisture flux in the extratropics during winter (Zhu and Newell, 1998). However, AR-induced meridional LHT significantly decreases as detector consensus increases. We demonstrate the extent of this change in MERRA2 reanalysis and a High-Resolution Model Intercomparison Project (HighResMIP) model (Haarsma et al., 2016). Our AR detector uniquely enables exploration of AR-induced LHT in HighResMIP, since there currently exists no ARTMIP HighResMIP experiment. Due to its increased resolution, HighResMIP offers a valuable testbed to explore precipitation and the hydrological cycle.

In this manuscript, we present the following advances in AR detector uncertainty, CNN interpretability, and CNN generalizability:

1. We train ARCNNs to replicate the mean output of multiple AR detection algorithms in Sections 2.1-2.3.

2. Using style transfer, we train ARCNNs to generalize to a variety of input fields, resolutions, and datasets in Section 2.4. Crucially, this generalization does not require additional training labels.

3. We rigorously validate ARCNNs on an idealized climate dataset, in which the correct answer is known ahead of time in Section 3.

4. In Section 4, we use ARCNNs to assess the relationship between ARs and poleward LHT while considering AR detection uncertainty.



**Table 1.** Description of ARCNN experiments.

| | Labeled Training Datasets | Unlabeled Training Datasets | Loss Function | Generalization Capability |
|---|---|---|---|---|
| **Experiment 1** | MERRA2 IWV MERRA2 IVT | None | Perceptual Loss | Input Fields |
| **Experiment 2** | GRIDSAT Brightness Temperatures | None | Perceptual Loss | Input Fields |
| **Experiment 3** | MERRA2 IVT ERA20th Century IVT | None | Perceptual Loss | Resolutions |
| **Experiment 4** | MERRA2 IVT ERA-I IVT | None | Perceptual Loss | Resolutions |
| **Experiment 5** | MERRA2 IVT ERA-I IVT | CAM5 IVT | Semi-supervised Style Transfer | Labeled and Unlabeled Datasets |
| **Experiment 6** | MERRA2 IVT ERA-I IVT | ECMWF-IFS-HR IVT | Semi-supervised Style Transfer | Labeled and Unlabeled Datasets |

## 2   Training neural networks to detect atmospheric rivers

We train ARCNNs to detect ARs in ERA 20th Century Reanalysis (1° degree horizontal resolution) (Poli et al., 2016), MERRA2 reanalysis (0.5° x 0.625°) (Gelaro et al., 2017), ERA-I reanalysis (approximately 0.28°) (Dee et al., 2011), GRID-SAT (gridded satellite data, coarse-grained to 0.28° degrees) (Knapp et al., 2011), C20C CAM5 (0.25°) (Stone et al., 2019), and the ECMWF-IFS-HR HighResMIP climate model (0.5°) (Roberts et al., 2018). In CAM5 and ECMWF-IFS-HR, we use historical forcings; the simulations for these two models are named All-Hist and highresSST-present, respectively. In MERRA2, we detect ARs in both IWV and IVT; in ECMWF-IFS-HR and ERA-I we detect ARs in IVT; and in GRIDSAT, we detect ARs in the brightness temperatures in the infrared window. We select these datasets because they allow us to detect ARs in a variety of different reanalysis products, models, input fields, and resolutions. Using these datasets, we examine a truly generalizable framework for detecting ARs. Satellite data could enable real-time meteorological analysis of global AR risks, ERA20th Century Reanalysis includes information about long-term trends from 1900-present, high-resolution climate reanalyses enable study of localized AR impacts, and climate models can be used to study future changes in ARs. We summarize the ARCNNs we train in Table 1, and we explain the choice of loss function in Section 2.3 and 2.4.

### 2.1   The AR Consensus Index (ARCI)

To systematically assess AR detection uncertainty, the Atmospheric River Tracking Method Intercomparison Project (ART-MIP) (Shields et al., 2018; Collow et al., 2022; O'Brien et al., 2022) is a campaign to compare a variety of AR detection algorithms across a common set of reanalyses and climate simulations. For AR identification, these algorithms prescribe



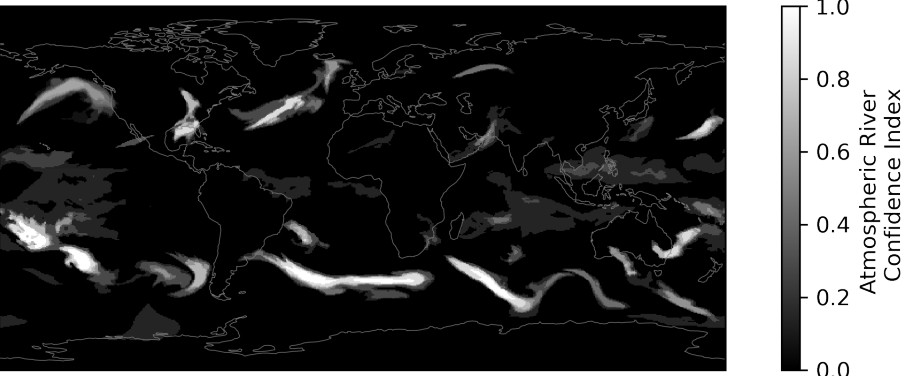

**Figure 3. Example of the Atmospheric River Consensus Index (ARCI).** ARCI (black and white shading) refers to the proportion of 14 AR detection algorithms that identify an AR. The ARCI ranges from 0 (none of the AR detection algorithms identified an AR) to 1 (all algorithms identified an AR). ARCI at the same time step as 1.

thresholds in Integrated Vapor Transport (IVT), Integrated Water Vapor (IWV), length-width ratio, and/ or minimum area. Xu et al. (2020) propose using heuristics from image processing to detect ARs without requiring thresholds in Integrated Water 110 Vapor and Integrated Vapor Transport; however, they still use thresholds in minimum area, maximum area, and length-width ratio.

In the ARTMIP Tier 1 experiment, the algorithms detect ARs in three-hourly MERRA2.0 reanalysis (Global Modeling And Assimilation Office and Pawson, 2015; Rutz et al., 2019). Using the detections from 14 ARTMIP algorithms, we assemble the *Atmospheric River Consensus Index (ARCI)*. We list the algorithms in Appendix D. ARCI represents the fraction of ARTMIP 115 algorithms that identify a specific horizontal grid cell and time step as falling within an AR. It is obtained by averaging the binary maps output from each algorithm (1 if the grid cell is in an AR, 0 otherwise). In this way, *ARCI is a measure of consensus among ARTMIP algorithms.*

In Figure 3, we visualize the ARCI at a sample time step. ARs emerge as long, narrow columns of moisture transport. To resolve the spread among ARTMIP algorithms, one solution could be a simple majority vote. However, many of the ARTMIP 120 algorithms are correlated, as they use similar thresholds in IWV, IVT, or area size. Therefore, detections from different ARTMIP algorithms are not independent, and a majority vote for AR detection may result in a misleading, statistically unsound picture of ARs. Since some ARTMIP algorithms are only designed for specific regions, the ARCI is defined to represent the proportion of algorithms that were run on a given grid cell. Some detection algorithms use latitude bounds, such that these features appear primarily in the midlatitudes. While some detection algorithms do identify ARs in the tropics, most algorithms treat them as 125 extratropical phenomena associated with poleward moisture transport (Zhu and Newell, 1998; Guan and Waliser, 2015; Nash et al., 2018; Newman et al., 2012) and midlatitude dynamics (Lora et al., 2020; Zhang et al., 2019; Gimeno et al., 2014; Ralph et al., 2018; Dacre et al., 2015).



## 2.2 The Neural Network

We train an ARCNN to identify ARs, using the ARCI dataset as training labels. Since these labels include a measure of
AR confidence, the ARCNN preserves an estimate of detector uncertainty in its AR detection. This is a unique benefit of
our training setup. Because the CNN is making a prediction for each grid cell, this learning objective is known as semantic
segmentation.[1]

We use the DeepLabv3+ CNN architecture (Chen et al., 2018), implemented in PyTorch (Paszke et al., 2019), because of
its strong performance in semantic segmentation tasks on several computer vision datasets. Additionally, this architecture is
explicitly designed for learning at multiple scales. It uses atrous convolutions, in which convolution filters are applied over fields
of view with multiple sizes. In computer vision, Chen et al. (2017) introduce this method to detect multiscale information in a
dataset of vehicles, household objects, and animals. We speculate that atrous convolutions make this architecture well-suited
for AR detection in climate datasets with different spatial resolutions.

For the CAM5 experiment (Experiment 5 in Table 1), we train on the 6th ensemble member (run006), from 1995-2005.
Due to memory limitations, we test on the year 1995 of the 8th ensemble member (run008); this year was chosen arbitrarily.
We are training and testing on different ensemble members, which diverge due to sensitive dependence on initial conditions.
Therefore, the test dataset is a proper out-of-sample evaluation of the ARCNN. Regarding all the other experiments in Table 1,
the years 1980-2019 were available for the MERRA2 dataset, 2006-2017 were available for ERA-I and GRIDSAT, 1950-2014
were available for ECMWF-IFS-HR, and 1980-2010 were available for ERA20th Century Reanalysis. The validation dataset
is composed of the years 1982, 1993, 2004, 2009, 2011; the test dataset is composed of the years: 1984, 1997, 2003, 2010,
and 2015 (if available). These years were chosen arbitrarily. For each dataset, the remaining years were used for training. The
validation and test years are interspersed with the training years to account for possible changes in the characteristics of ARs
during the historical record. In machine learning, the validation and test datasets are meant to be out-of-sample tests for the
CNN. Since atmospheric datasets are highly correlated on time scales of $\mathcal{O}(10)$ days, our training, validation, and test sets are
in yearly blocks rather than being selected randomly among all available time steps. If we chose the train set, test set, and
validation set by random shuffling of all time steps, then elements of the latter two sets would be highly correlated with the
training data. This would corrupt the validation and test datasets, and they would no longer be an out-of-sample test for the
neural network. Using random shuffling could lead to artificially high performance on the validation and test sets because they
would not be reliable indicators of the CNN's performance.

Two stages of the machine learning pipeline are training and inference. During training, the CNN's parameters are optimized
to learn the relationship between the input and the ARCI labels. With a batch size of 2, we train the each ARCNN using 1
NVIDIA GTX 1080 Ti GPU with 11 gigabytes of memory. For each ARCNN, training takes roughly 24 hours. In machine
learning, the training dataset is often augmented and perturbed to synthetically increase the size of the training dataset (Wong
et al., 2016). During training, we augment the input to the ARCNNs: with 50% probability, we multiply the input by a random

---

[1]Other learning objectives include classification, where the CNN makes one prediction for the entire input field, and bounding box detection, where the
CNN learns to draw bounding boxes around objects of interest in an image. Liu et al. (2016) classify cropped regions of global simulations as AR or not-AR,
and Racah et al. (2017) draw bounding boxes around ARs in global input fields.



factor between 0.92 and 1.08, and with 50% probability, we uniformly add a constant to the input field. The constant is
sampled uniformly between -15 mm and 15 mm for IWV, between -175 and 175 kg/m/s for IVT, and between -9 and 9 K for
for GRIDSAT brightness temperatures. As an additional form of data augmentation, we degrade the resolution of the input
field by a factor of 2 or a factor of 4, with 33% probability for each. The remainder of the time during training, we do not
degrade the resolution.

During inference, the CNN's weights are frozen, and the CNN can generate AR detections. For inference, we integrate the
CNN into an open-source package called the Toolkit for Extreme Climate Analysis (TECA v4.0.0) (Prabhat et al., 2015, 2012;
Burlen Loring et al., 2022)[2]. With TECA, the CNN can be deployed on large climate datasets in parallel and can efficiently
leverage high-performance computing. Using 1484 nodes on the Cori supercomputer at NERSC, TECA generated AR detec-
tions in 280GB of ECMWF-IFS-HR IVT. Each node uses Intel's Xeon Phi "Knight's Landing" processor, with 68 cores and
96GB per node.

### 2.3  Extending the neural networks to different input fields and resolutions

Using an optimization algorithm and a training dataset, CNNs approximate a function that transforms the input field into the
labels. A key strength of machine learning algorithms is that they do not rely on an "explicitly programmed" relationship
between the input and the labels (Arthur Samuel, 1959); this relationship is learned from data. On the other hand, ARTMIP
algorithms do use explicit programming: experts manually define and implement a set of heuristics and thresholds to detect
ARs. In this study, the majority of the ARTMIP algorithms rely on IVT thresholds for AR identification. Additionally, in the
ARCI dataset we create, all the ARTMIP algorithms are tuned and run on MERRA2. We use CNNs to replicate ARCI in
different input fields and historical datasets: 3-hourly 0.5° x 0.625° MERRA2 IWV, 3-hourly 0.25° ERA-I IVT (Dee et al.,
2011), 3-hourly 1° ERA-20th Century Reanalysis IVT (Poli et al., 2016), and the 3-hourly brightness temperatures in the
infrared window from gridded satellite data (GRIDSAT), which is coarse-grained to ERA-I resolution (Knapp et al., 2011).
Our goal is to use the labels from existing heuristic algorithms to detect ARs in a variety of resolutions and fields.

In this section, we train the neural network using the ARCI dataset as the labels. For each ARCNN, we use a variety
of datasets as possible input fields, summarized in Table 1. We use bilinear interpolation to regrid the ARCI labels to the
resolution of the input field. CNNs are powerful function approximators. In Experiment 1 in Table 1, we train *one* ARCNN to
detect ARs in either IWV or IVT. We trained our CNN by selecting at random with equal probability either the IVT or IWV
field as input in a given training pass. We visualize the detections in IWV in Figure 4. Despite the major differences between
these fields, the CNN reliably detects ARs regardless of which field is used as input.

In Experiment 2 of Table 1, we explore AR detection in satellite images. Currently there exist only a few methods of
detecting ARs in satellite observations, and these methods do not leverage the wind fields or moisture transport for their
detection (Neiman et al., 2008; Ralph et al., 2004). Our training setup with CNNs enables replication of the full ensemble of
ARTMIP algorithms to infrared satellite observations. In Figure 4, we illustrate the ARCNN's detections in GRIDSAT.

---

[2]https://github.com/LBL-EESA/TECA



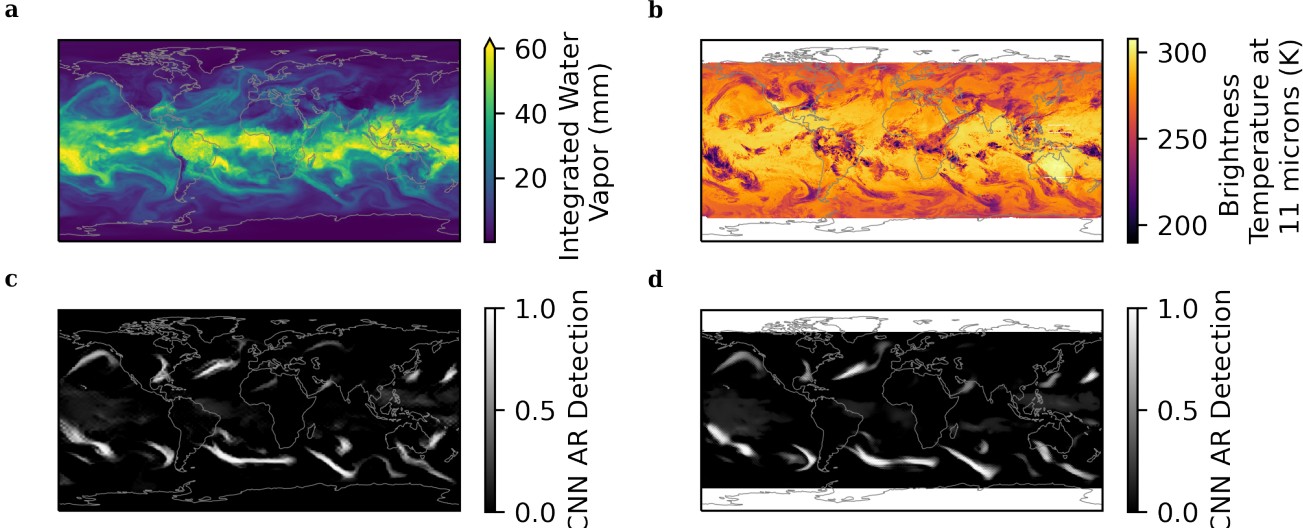

**Figure 4. CNN-based AR detections in MERRA2 IWV and GRIDSAT predictions. (a)** MERRA2 IWV and the **(b)** Infrared Window of GRIDSAT at the same time step as Figures 3 and 1. **(c)** CNN AR detections from MERRA2 IWV and **(d)** CNN AR Detections from GRIDSAT.

Through the Adam optimization algorithm (Kingma and Ba, 2014) with a learning rate of $10^{-5}$, the CNN finds the optimal set of weights that minimize a loss function, $\mathcal{L}$. $\mathcal{L}$ quantifies the difference between the DeepLabv3+ predictions and the ARTMIP labels. To train a CNN that can reliably detect ARs in multiple resolutions, we explore different loss functions. One option for $\mathcal{L}$ is a pixelwise cross entropy:

$$\mathcal{L} = \sum_{i=1}^{k} y_i \log(\hat{y}_i)$$

where $k$ is the number of grid cells, $y$ is the label (ARCI), and $\hat{y}$ is the CNN AR detection. This is a *pixelwise* loss function because the prediction is compared to the label at each grid cell, or "pixel." Johnson et al. (2016) demonstrate that a pixelwise loss function often yields subpar results. Instead, they propose a *perceptual* loss function. Rather than comparing the predictions to the AR labels grid cell by grid cell, a perceptual loss function compares a low-dimensional representation of the labels to a low-dimensional representation of the CNNs' detections. This loss function is less noisy and susceptible to changes in individual pixels. To justify perceptual loss functions, Johnson et al. (2016) use the following analogy: if $\hat{y}$ was shifted by 1 pixel from $y$, then the pixelwise cross entropy loss would be considerable, but a perceptual loss function would be low.

To extract the low-dimensional representation from the labels and predictions, the perceptual loss function itself utilizes another CNN. We call this neural network the Loss Neural Network, and we refer to the low-dimensional representation extracted by the Loss Neural Network as *features*. Thus, perceptual loss functions involve two separate neural networks: (1) the ARCNN and (2) the Loss Neural Network. The ARCNN is the DeepLabv3+ CNN being trained to detect ARs; its weights





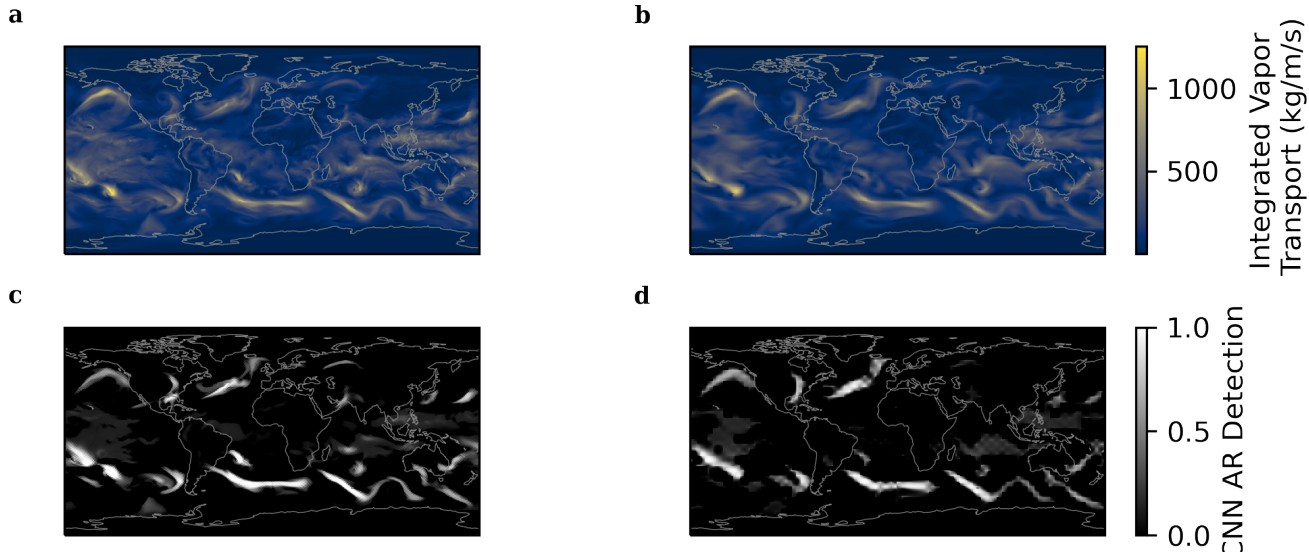

**Figure 5. CNN-based AR detections in ERA-I IVT and ERA-20th Century Reanalysis.** (Top Left) ERA-I IVT and the ERA-20th Century Reanalysis IVT (Top Right) at the same time step as Figures 3 and 1. (Bottom Left) CNN AR detections from ERA-I IVT and (Bottom Right) CNN AR Detections from ERA-20th Century Reanalysis. ERA-I has a 0.25° x 0.25° spatial resolution, while ERA-20th Century Reanalysis has 1° x 1° resolution.

.

are being updated to minimize the loss function. The Loss Neural Network has frozen weights; it is only used to extract a low-dimensional representation from the predictions and labels. The Loss Neural Network is a general feature extractor, pre-trained on a large dataset of images called ImageNet (Deng et al., 2009), and it uses a standard architecture, called VGGNet
(Simonyan and Zisserman, 2014).

Let $\phi$ denote the features extracted by the Loss Neural Network, and $y$ denote the ARCI labels. $\phi(y)$ has the shape $C \times HW$, where $C$ denotes the number of convolutional filters in that layer of the architecture, $H$ denotes the height of the features, and $W$ denotes the width of the features. $H$ and $W$ are determined by the architecture of the Loss Neural Network and the resolution of the input. $H$ and $W$ correspond to the dimensions of the compressed representation of the input. A perceptual
loss function is composed of two components:

$$\mathcal{L} = \mathcal{L}_{content} + \mathcal{L}_{style}$$

Content loss, $\mathcal{L}_{content}$, is designed to encourage the AR Detection Network to correctly locate ARs. Content loss is the squared norm between the features of the predictions and the truth.

$$\mathcal{L}_{content} = ||\phi(y) - \phi(\hat{y})||^2$$





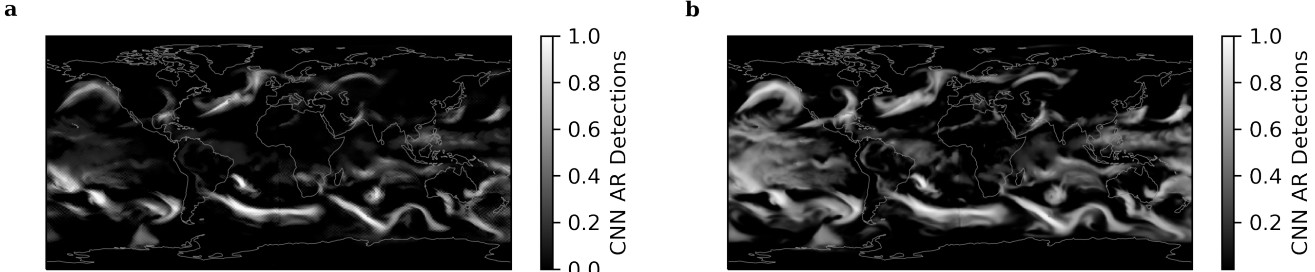

**Figure 6. CNN-based AR Detections with and without Perceptual Loss.** CNN AR detections in 0.25° ERA-I at the same time as Figures 1-4 **(a)** with a perceptual loss function and **(b)** with a pixelwise cross-entropy loss function.

where $\hat{y}$ refers to the ARCNN's predictions.

Style loss, $\mathcal{L}_{style}$, is designed to ensure that the predictions of the AR Detection Network have the correct stylistic char-

acteristics as the labels. These characteristics are represented by a Gram matrix $G$. where $G(y) = \phi(y)\phi(y)^T$. According to Johnson et al. (2016), the Gram matrix can be interpreted as the $C \times C$ covariance matrix between the $C$-dimensional features extracted by the Loss Neural Network. Style loss minimizes this covariance of the labels and of the prediction:

$$\mathcal{L}_{style} = ||G(y) - G(\hat{y})||^2$$

$\mathcal{L}_{style}$ operates on Gram matrices, which always have the shape $C \times C$ regardless of the resolution of the input. The value of $C$ is determined by the architecture of the Loss Neural Network, VGGNet. On the other hand, $\mathcal{L}_{content}$ operates on $\phi$, which

has shape $C \times HW$. $H$ and $W$ change based on the resolution of the input. Similarly, a pixelwise loss function relies on the resolution of the input dataset, since each grid cell in the prediction is compared to each grid cell of the input. As in Section 4 of Johnson et al. (2016), we calculate style loss using the features extracted from the first four layers of the Loss Neural Network, and we calculate feature reconstruction loss from the second layer of the Loss Neural Network

Since $\mathcal{L}_{style}$ is independent of the input resolution, it provides the flexibility to train a CNN to detect ARs across resolutions.

In Figure 5, we visualize the predictions in ERA-20th Century Reanalysis and ERA-I at the same sample time step. One ARCNN was trained on MERRA2 and ERA-20th Century Reanalysis (Experiment 3 in Table 1), and the other ARCNN was trained on MERRA2 and ERA-I (Experiment 4). These ARCNNs have learned to detect ARs in input datasets with different resolutions, as their AR detections match the labels from the ARCI dataset (Figure 3). In Figure 6, we demonstrate the utility of perceptual loss functions. With a pixelwise loss function, a CNN trained to predict ARs in MERRA2 (0.5 degree

horizontal resolution) and ERA-I (0.25 degree horizontal resolution) greatly overpredicts AR probabilities in the latter. This overprediction occurs despite the fact that both datasets are used in training: the pixelwise loss function does not enable effective learning across resolutions. For learning across resolutions, the pixelwise loss function may have subpar performance because different resolutions have different numbers of grid cells (aka pixels for the CNN). The detections from a CNN trained with a pixelwise loss function (Figure 6b) are consistently larger than those from ARCI (Figure 3). In particular, the CNN in





Figure 6b overpredicts ARs in the tropical Pacific, and the midlatitude ARs have a larger area compared to Figure 3. Using a perceptual loss function, the detections (Figure 6a) are very similar to those in Figure 3. The detected ARs have a similar spatial extent and covariance to the ARs from ARCI. Thus, a perceptual loss function enables an ARCNN to detect comparable ARs in the MERRA2 and ERA-I datasets despite their different spatial resolutions. We note that the ARCNN was trained with both MERRA2 and ERA-I. It has not learned to generalize to a new dataset that it has not seen in training; rather, the perceptual loss function enables the ARCNN to learn most effectively from the different resolutions in the training dataset.

One artifact of using the perceptual loss function is that an unphysical checkerboard pattern appears in the detected ARs. This pattern is a known byproduct of CNNs' deconvolutional layers, which convert features (which are a low-dimensional representation of the input) into the predictions (which have a higher resolution than the input) (Aitken et al., 2017; Odena et al., 2016). Additionally, the pattern could also be caused by the fact that the Loss Neural Network was trained on ImageNet, in which all images have the same resolution. Further research is necessary to characterize the exact origin of this pattern and to see if a Loss Neural Network trained on high-resolution images can ameliorate this problem. The pattern is most visible in Figure 5d in detections in ERA 20th Century Reanalysis, but it is also visible in higher resolution AR detections (Figures 4c, 4d, and 6a). Because the loss function is crucial to enabling detection across datasets, we conclude that its benefits outweigh the disadvantage of the introduced pattern. In particular, the checkerboard variations are roughly an order of magnitude smaller than the AR detections themselves, so the detected ARs are still clearly discernible. The ARCNNs trained with perceptual loss functions are extensively validated using a semantic segmentation metric (Section 2.5), an idealized model (Section 3), and a comparison of AR-induced heat transport in ARCNN predictions and ARCI (Section 4.

## 2.4 Extending the neural networks to climate models using style transfer

When applying neural networks to climate datasets, a central challenge concerns generalization to new climate datasets and scenarios (Reichstein et al., 2019; Beucler et al., 2021). CNNs are exceptionally good at interpolation, in which they are applied to scenarios similar to the training dataset, but they are poor at extrapolating to new scenarios. To address this challenge, we use style transfer to generalize the ARCNNs. Our goal is to train ARCNNs to reliably detect ARs in climate reanalyses and climate models without requiring time-intensive intercomparisons for each new climate dataset. We use style transfer to extend the ARCNN to climate simulations in which the ARCI dataset does not have a labeling campaign. In this way, style transfer enables the CNNs to be highly transferable across datasets; it enables training on datasets with labels (historical reanalyses) and datasets without labels (free-running climate simulations). As in the previous section, style transfer still requires training on input from unlabeled climate simulation: it does not enable an ARCNN to detect ARs on a dataset entirely unseen during training. Using style transfer, we train a CNN to detect ARs in the European Center for Medium-range Weather Forecasts climate simulation (Roberts et al., 2018) for the High-Resolution Model Intercomparison Project (HighResMIP) (Haarsma et al., 2016) and 0.25° C20C simulation with the Community Atmosphere Model v5 (Neale et al., 2010; Stone et al., 2019). These refer to Experiments 5 and 6 in Table 1.

This problem formulation is a form of semi-supervised learning, in which the CNN must learn to detect ARs in both labeled datasets (MERRA2) and unlabeled datasets (ECMWF-LS-HR or C20C CAM5). On the other hand, in supervised learning, a





CNN learns to detect ARs by only learning from labeled data. Let $D_l$ refer to a labeled dataset (e.g. MERRA2), and let $D_u$
refer to an unlabeled dataset. A supervised perceptual loss function would be

$$\mathcal{L}_{\text{supervised}} = \mathcal{L}_{\text{style}}^{D_l\ preds, label} + \mathcal{L}_{\text{content}}^{D_l\ preds, label}$$

where the superscripts denote the datasets on which $\mathcal{L}$ is calculated. This supervised loss function just calculates the style
loss and content loss on a labeled dataset. With this supervised loss function, a supervised model trained on ERA-I does
not generalize well to ECMWF-IFS-HR due to the fundamental differences between climate models and reanalysis. Free-
running climate models rely on numerical simulation of the atmosphere and ocean, while reanalysis incorporates observations
and data assimilation from in-situ measurements. Therefore, a neural network trained on the reanalysis cannot generalize or
extrapolate to a climate simulation. This can be seen in Figure 7. The CNN's detections on ECMWF-IFS-HR (Figure 7d) are
significantly lower than the ARCI labels on MERRA2 (Figure 7b). While ECMWF-IFS-HR is fundamentally a different data
product than MERRA2, it is reasonable to expect that ARs in ECMWF-IFS-HR should be relatively common, even if the exact
spatiotemporal pattern doesn't exactly match in these two datasets. However, there is an order of magnitude difference between
AR detections in the two datasets. This suggests that the AR detector is not generalizing to a new dataset. Additionally, the
CNN does not appear to be underpredicting ARs in a spatially uniform manner. In the Northern Hemisphere, there appears to
be a markedly higher probability of ARs over the Atlantic than the Pacific, but this difference does not exist in the ARCI labels
of Figure 7b.

To address this lack of generalization, we convert the loss function from a supervised loss to a semi-supervised loss. A
supervised loss only learns from labeled data, but a semi-supervised loss learns from both labeled and unlabeled data. We
add another term to the loss function: a style loss between the MERRA2 ARCI labels and the ARCNNs' predictions on an
unlabeled dataset.

$$\mathcal{L}_{\text{semi-supervised}} = \mathcal{L}_{style}^{D_l\ preds, D_l label} + \mathcal{L}_{content}^{D_l\ preds, D_l label} + \mathcal{L}_{style}^{D_u\ preds, D_l label}$$

The additional loss term, $\mathcal{L}_{style}^{D_u\ preds, D_l label}$, represents the style loss between the predictions on the unlabeled dataset and
the labels on the labeled dataset. By definition, on an unlabeled dataset, there are no labels to learn from. To overcome this
challenge, in unlabeled datasets the semi-supervised loss encourages the network to make predictions that have similar feature
Gram matrices as those of ARCI labels.

In Figure 7, we demonstrate that the semi-supervised training scheme improves the quality of AR detections in unlabeled
ECMWF-IFS-HR simulations. This corresponds to the ARCNN used in Experiment 6 in Table 1. With the supervised training
loss, the neural network is consistently underconfident (Figure 7d), and its detected AR probabilities are too low. However, with
a semi-supervised training scheme, the AR detections in EMCWF-IFS-HR (Figure 7c) have a similar magnitude and spatial
pattern to the ARCI labels (Figure 7b). Therefore, the semi-supervised style loss addresses this consistent underconfidence and
more accurately identifies ARs in both hemispheres. To summarize the training scheme presented in Sections 2.3 and 2.4, we
include a schematic diagram in Figure B1. This diagram displays the role of labeled and unlabeled datasets in the loss function.



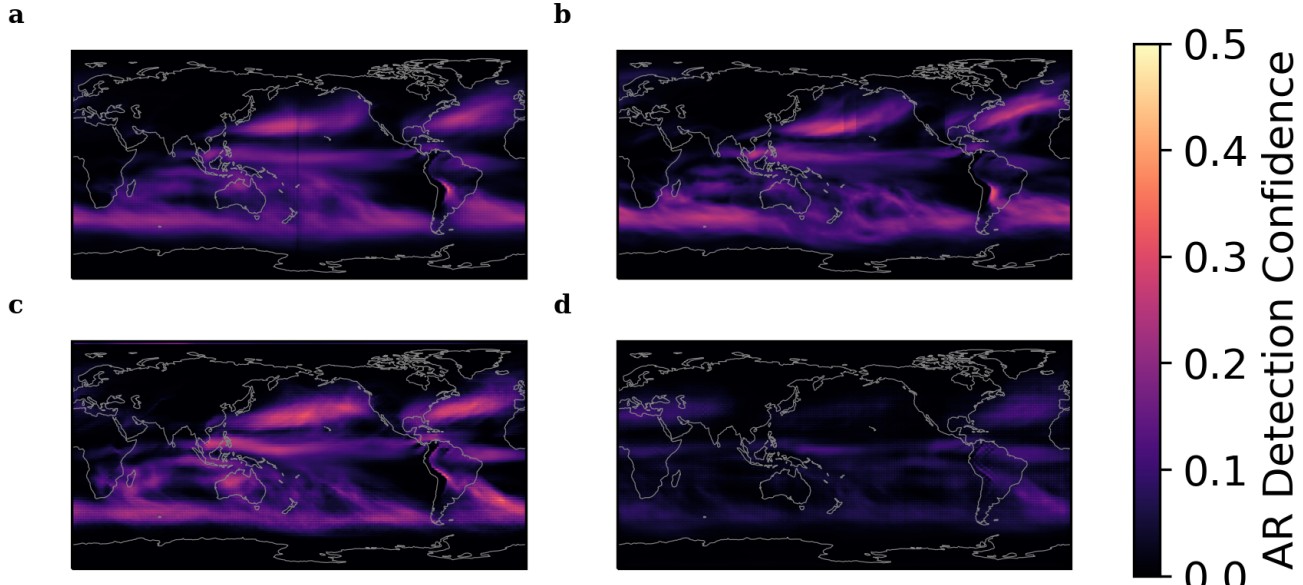

**Figure 7. DJF ARCNN Detections with and without Style Transfer. (a)** ARCNN detections in MERRA2. This ARCNN was trained to detect ARs in MERRA2 using either IWV or IVT as input. **(b)** ARCI labels on MERRA2. **(c)** ARCNN (trained with style transfer) in ECMWF-IFS-HR **(d)** ARCNN (trained without style transfer) detections in ECMWF-IFS-HR. **(a)-(d)** All figures show the DJF average during the years of the test dataset (1984, 1997, 2003, and 2010).

Using style transfer to generalize a CNN to new datasets is grounded in machine learning theory. Li et al. (2017) show

that style transfer is a form of domain adaptation, which "aims to transfer the model that is learned on the source domain to the unlabeled target domain." Their derivation shows that style transfer minimizes the Maximum Mean Discrepancy (MMD) between a sample in the source domain and a sample in the target domain. MMD is a metric that quantifies the difference between the distributions underlying the two samples. By minimizing the MMD, style transfer aligns the feature distributions of the ARCNN predictions in the source and target domains. This is precisely our goal for using style transfer: we want to

ensure that AR detections in the source domain (MERRA2) have the same feature distribution as those in the target domain (historical climate simulations). Style transfer enables offers a computationally efficient way to integrate this distribution alignment directly into the training of the ARCNN. In the field of computer vision, Atapour-Abarghouei and Breckon (2018) use style transfer for a similar purpose: they generalize a neural network across synthetic data and real-world data.

We apply style transfer and the resulting distribution alignment to historical reanalysis and the historical scenario of climate

model simulations. We do not use style transfer on ARs in future climate simulations with different emissions scenarios. This would artificially constrain AR detections to have the same Gram matrices in historical and future datasets. This behavior is not necessarily accurate, as O'Brien et al. (2022) find notable changes to future intensity and size of ARs. In order to explore





future changes in ARs, an ARCNN trained with style transfer on a historical simulation could be applied to a future simulation. Then, changes in AR frequency and intensity between the historical and future simulations could be explored.

## 2.5 Assessing the performance and robustness of neural networks

Using the experiment setup described above, we train six different ARCNNs. Each is trained with a different input dataset, field, or combination thereof. We evaluate the ARCNNs' performance using the Intersection over Union (IoU) score. IoU is calculated as the intersection between the ARCNN predictions and the ARCI labels divided by the union of those two quantities. For calculation of this metric, we binarize ARCNN predictions and labels at 0.67. This metric ranges from 0 (worst)

to 1 (best). We refer the reader to Prabhat et al. (2021) for an in-depth discussion and visualization of the IoU score between CNN predictions and labels. The IoU metric is commonly used to evaluate CNNs for semantic segmentation (Chen et al., 2018), and it has been used in prior work on CNN-based AR detection (Prabhat et al., 2021; Kurth et al., 2018). In Figure 8, we show the IoU scores for ARCNNs trained and evaluated on different combinations of climate datasets. The x axis of Figure 8 corresponds to Experiments 1-5 in Table 1. In all cases, the ARCNNs' IoU scores surpass 0.65, and as shown by the sample

outputs (Figures 4-7), the ARCNN predictions closely match those of the ARCI labels (see Figure 3 for an instantaneous snapshot and Figure 7a for a time mean).

In addition, we show that the ARCNNs' performance is robust to perturbations of the input field. We perturb the input field by multiplying it by factors ranging from 0.9 to 1.1 in increments of 0.02. For each perturbation, we calculate the ARCNN's mean IoU score across all samples on the test dataset, resulting in eleven IoU scores. In Figure 8, the spread of these eleven

IoU scores, indicated by the box and whiskers, is relatively small for each CNN. Despite these perturbations, each neural network's IoU score remains almost constant. This is an important quality for an AR detector, and existing work has found that for threshold-based AR detection algorithms, AR area can be sensitive to changes in parameters (Newman et al., 2012), and Reid et al. (2020) show how AR count changes as a function of different thresholds in IVT.

Figure 8 illustrates neural networks' ability to probabilistically detect ARs across fields, resolutions, and datasets. First,

Figure 8 demonstrates that neural networks can detect ARs across fields. We train one neural network, labeled "MERRA2 IWV, IVT", to detect ARs regardless of the input field (IWV or IVT): this network is Experiment 1 in Table 1. During training, this CNN sees 50% of its training set as IWV data and 50% as IVT data. This CNN's outputs were discussed and visualized earlier, in Section 2.3 and Figure 4c. The high IoU scores, between 0.75 and 0.8, illustrate the ability of the CNN to detect ARs across fields.

Many of the AR detection algorithms submitted to ARTMIP require the IVT field, as this field can readily be subjected to a threshold for AR detection. This field is not routinely saved in climate model simulations, so its calculation requires substantial disk storage and RAM to calculate the required integral from the 3-dimensional wind and water vapor fields. Unlike IWV and GRIDSAT, calculation of IVT requires zonal and meridional wind at all pressure levels and times. We can successfully train neural networks to detect ARs in fields such as IWV and GRIDSAT (Figure 4). Therefore, we conclude that even though these

fields lack information regarding atmospheric dynamics, they still contain the core identifiable properties of an AR. Figure 8 shows that CNNs can still reliably detect ARs in IVT, IWV, and GRIDSAT.



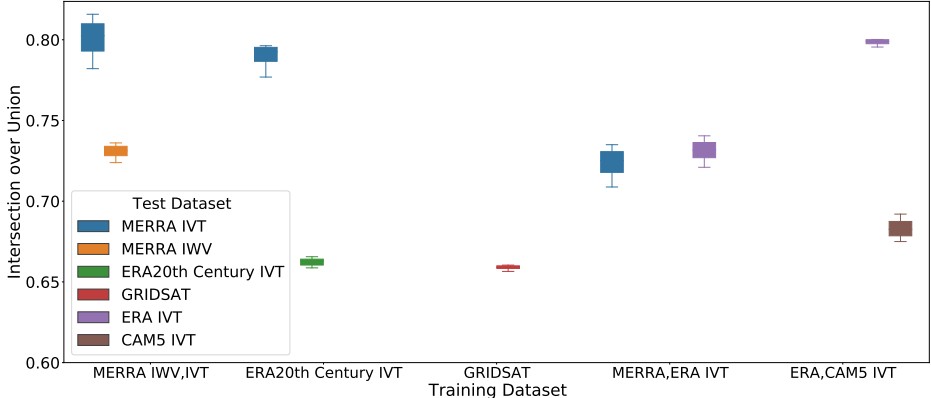

**Figure 8. Performance of ARCNNs on different climate datasets and input fields**. Five ARCNNs are trained on different datasets and input fields, shown on the x-axis. Their performance is measured by Intersection over Union scores (IoU: 0 being the worst, 1 being the best) on different test datasets. On each test dataset, the input field is multiplied by a factor ranging from 0.9 to 1.1 in increments of 0.02, and the IoU score is calculated. For the resulting eleven IoU scores, the box shows the lower and upper quartile, and the whiskers show the mininum and maximum of the IoU scores. All IoU scores are on test set years: 1984, 1997, 2003, 2010, and 2015.

Second, Figure 8 demonstrates networks can detect ARs across resolutions. A network trained on MERRA2 and ERA-I can successfully identify ARs across both datasets' resolutions with an IoU score of approximately 0.72. This represents an important advantage of our training schema. Collow et al. (2022) show that a prior CNN trained without semi-supervised style transfer has significant differences when run on different reanalyses.


Finally, Figure 8 demonstrates that our experimental setup can train CNNs that detect ARs across datasets. Using only MERRA2.0 AR labels as training data, the CNN successfully identifies ARs in CAM5 (Experiment 5 in Table 1) and ECMWF-IFS-HR (Experiment 6 in Table 1) by leveraging the semi-supervised perceptual loss function above. After training on unlabeled CAM5 as input, we validate the CNN's predictions on CAM5 AR detections obtained from the ARTMIP High Resolution Tier 2 (Shields et al., 2023); we validate on the year 1995 of the 8th ensemble member of CAM5 All-Hist. These CAM5 labels
are only used for validation of the trained neural network; they are not used for training. Figure 8 shows the results for five trained ARCNNs (Experiments 1-5 in Table 1). A sixth ARCNN is trained using style transfer to detect ARs in ECMWF-IFS-HR (Experiment 6 in Table 1). Since there are no ARTMIP catalogs available for this dataset, its IoU score cannot be calculated. However, the time-mean spatial patterns of the detected ARs broadly match those of the ARCI labels, as discussed
in Section 2.4.

By detecting ARs in different climate datasets and fields, the ARCNNs cannot have learned to simply replicate the threshold-based detection algorithms on which they were trained. For instance, a threshold-based algorithm in IVT will not generalize to IWV and vice versa because these quantities represent different quantities with different magnitudes and spatial statistics. However, these ARCNNs are more generalizable AR detectors that have learned to recognize AR patterns in multiple fields.



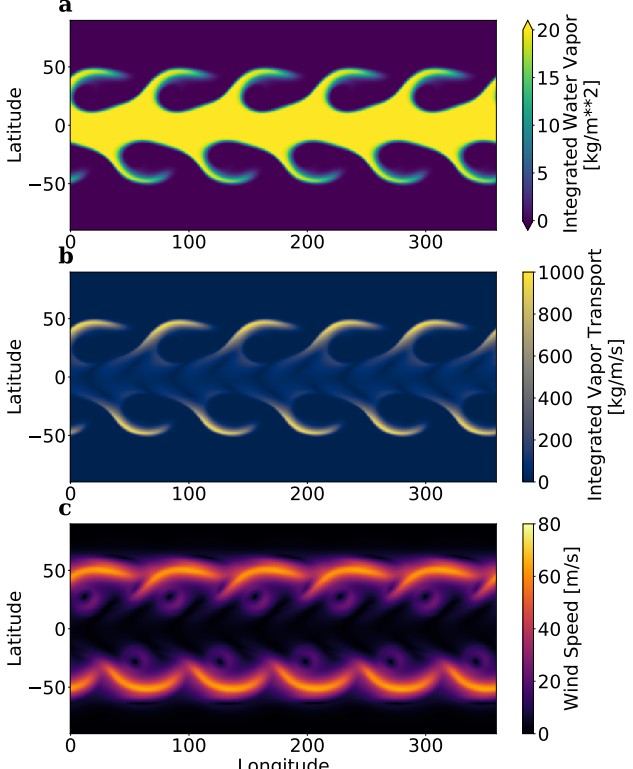

**Figure 9. Snapshot of the idealized simulation. (a)** Integrated Water Vapor, **(b)** Integrated Vapor Transport, and **(c)** Wind Speed are visualized. In this single layer idealized simulation, water vapor is a passive tracer. Rossby waves in the midlatitude jet streams lead to the formation of AR-like filamentary structures, which are responsible for all the water vapor transport outside the tropics.

## 3   An idealized dynamical test to validate the neural network

In Section 2.5, we assess the robustness of ARCNNs by comparing them to ARCI labels. However, since they rely on uncertain heuristic algorithms, they do not serve as an unambiguous ground truth of AR detections. Additionally, CNNs have millions of tunable parameters, which poses a significant challenge for their interpretability and trustworthiness. For these two reasons, it is vital to validate that ARCNNs make the right detections for the right reasons. We test the ARCNNs using an idealized model, where the correct AR detections can be rigorously determined. We design an experiment in which ARs should be responsible for the overwhelming majority of vapor transport, in both the zonal and meridional directions.

We test the CNN on AR-like filamentary structures in an idealized climate simulation based upon a dry, single-layer model. This is a simulation of shallow water layer with a basic state depth of 10,000 meters. The simulation is initialized with a barotropically unstable jet stream, as defined in Galewsky et al. (2004), and a passive tracer in the tropics, between 22.5°N and 22.5°S. The Galewsky et al. (2004) initial condition uses perturbations to the geopotential height field to induce the barotropic instability. To create multiple AR-like structures, we initialize the height field with five perturbations (each de-





fined by Galewsky et al. (2004)) equidistant across the 45° latitude circle. To induce flow in both hemispheres, we mirror the Galewsky et al. (2004) initial condition about the equator, such that there is a barotropically unstable jet in both the Northern and Southern Hemispheres. We implement and run this simulation using the SHTns Python package (Schaeffer, 2013). The
governing equation of the simulation is conservation of the quasi-geostrophic potential vorticity (QGPV) in the QGPV equation. However, we relax the beta-plane assumption in the quasi-geostrophic equations, and we define the Coriolis parameter $f$ as $2\Omega\sin\theta$, where $\Omega$ is the rotation rate of the Earth and $\theta$ is latitude. The simulation is run at 0.625 x 0.625 degree horizontal resolution. The full simulation and simulation parameters can be run using a single Python file in our open-source code repository (see the Code and Data Availability section for more details). The full simulation uses implicit diffusion and a fourth-order
Runge-Kutta numerical scheme.

In our simulation, water vapor is a passive tracer, and there are no interactions between water vapor and the flow field. Due to the barotropic instability, we can unequivocally identify the formation of a Rossby wave in each jet stream. This wave generates turbulent stirring and filamentary structures that are qualitatively consistent with the Zhu and Newell (1998) (ZN98) definition of an AR (a long, narrow filament of moisture). In Figure 9, we include a snapshot after the AR-like features have formed.
IWV, IVT, and wind speed are shown in the snapshot, and an animation of the full simulation can be seen at the following link: https://youtube.com/watch?v=7Gq7e5PIRio. Due to the simulation's idealized nature, all the water vapor transport in the extratropics is due to the AR-like filamentary structures. In order to pass our test, we expect the ARCNN's identifications should be responsible for the vast majority of the vapor transport in the extratropics. In Figure 10, we test whether the ARCNN's identifications are consistent with this expectation. For this test, we use the "MERRA IWV, IVT" ARCNN, which is trained on
both IWV and IVT. Figure 10a shows the ARCNN predictions binarized at 0.5. (Because this simulation is highly idealized, we binarize the CNN predictions and do not consider detector uncertainty.) Figures 10a and 10b show that ARs account for virtually all the vapor transport in the midlatitudes. As shown in Figure 10b, at the latitude of peak transport in the Northern Hemisphere, ARs account for 95% of IVT, and at the latitude of peak transport in the Southern Hemisphere, ARs account for 98% of total transport. Additionally, the detections in Figure 10a identify the filamentary structures as ARs. Thus, the ARCNN
identifications are consistent with our expectations in this idealized dataset. By testing the CNN in a context in which we know the right answer, we have more confidence in its operations on more complex datasets, such as historical reanalysis and high-resolution climate models.

## 4 Poleward latent heat transport induced by atmospheric rivers

In Sections 2 and 3, we validated the ARCNN detections on ARCI labels and on an idealized simulation. In this section, we
use the generalizable CNN to study the role of ARs in the climate system: specifically, we explore the relationship between ARs and poleward latent heat transport. To further validate the ARCNN, we test that it has the same estimates of AR-induced poleward LHT as the ARCI dataset. Also, we calculate AR-induced LHT in ECMWF-IFS-HR, on which there are no ARCI labels available.





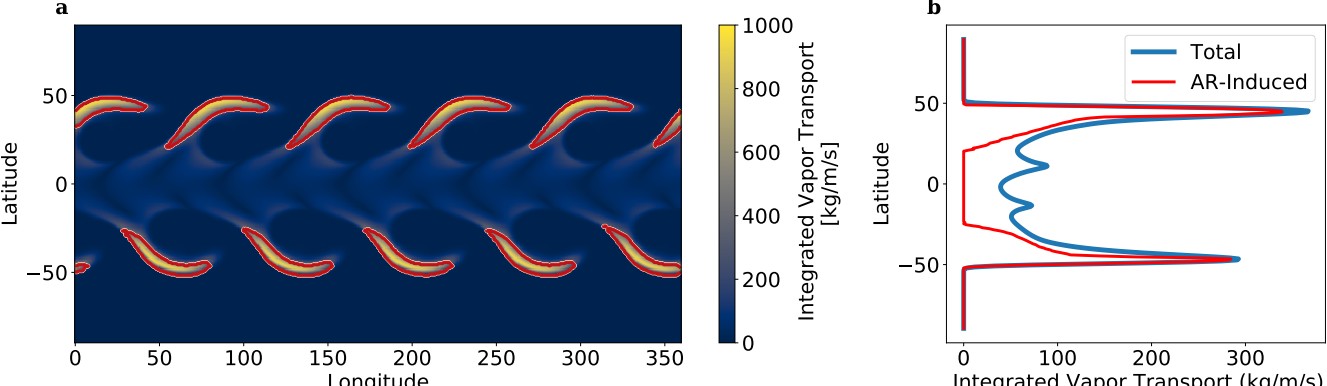

**Figure 10. Validating the CNN's AR Detections on an Idealized Simulation. (a)** Integrated Vapor Transport in the idealized simulation. The CNN's AR detections are shown with the red contours. **(b)** Zonal mean of (a), showing total IVT and AR-induced IVT.

At each latitude, there is an imbalance between absorbed solar radiation and outgoing longwave radiation (Hartmann, 2016).
This imbalance results in a surplus of energy input in the tropics and a deficit of energy input in the extratropics and polar regions. To maintain energy balance, energy is transported meridionally (in the north-south direction), and the net meridional transport is poleward in both hemispheres (Peixoto et al., 1992). This energy takes the form of latent heat, sensible heat, and geopotential energy. Here we investigate the relationship between ARs and meridional latent heat transport (LHT), which is defined as

$$
\text{LHT} = \frac{2\pi a \cos(\theta)}{g} \int\limits_{0}^{p_s} L_v [\overline{qv}] \, dp
\tag{1}
$$


$L_v$ is the latent heat of vaporization, $a$ is the radius of the Earth, and $\theta$ is latitude. The overline indicates the time mean (during the test set years), and the square brackets indicate the zonal mean.

In their original paper on ARs, Zhu and Newell (1998)(ZN98) find that ARs are responsible for more than ninety percent of meridional moisture flux in the extratropics. Using a newer detection algorithm with relative thresholds and geometric
constraints, Guan and Waliser (2015) and Nash et al. (2018) also reach a similar conclusion. Newman et al. (2012) define AR conditions as positive low-level wind anomalies and positive moisture anomalies. Under this definition, they find that ARs are the primary regions of extratropical moisture transport, confirming the claim put forth by ZN98. During winter, meridional LHT is dominated by transient eddies (Trenberth and Solomon, 1994), and since ARs are transient phenomena associated with extratropical cyclones and warm conveyor belts (Ralph et al., 2020), they are likely an important mechanism related to these
transient eddies.

Using Equation 1, we calculate total poleward LHT within the detected ARs. Shields et al. (2019) calculate LHT associated with landfalling ARs in western North America, the Iberian Peninsula, and the United Kingdom. We extend upon Shields et al. (2019) by assessing the role of all ARs, not just landfalling ones; this extension enables exploration of ARs' role in maintaining energy balance in the climate system.





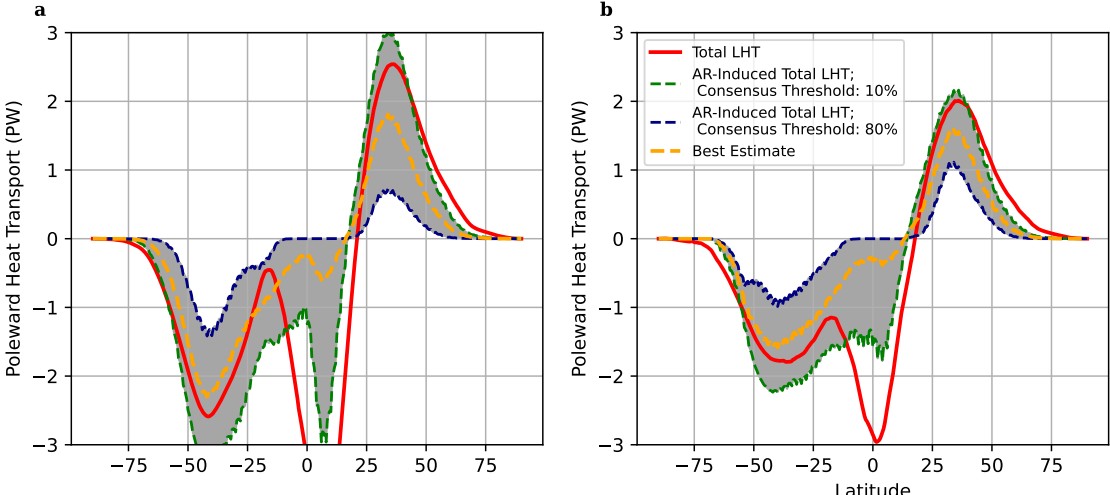

**Figure 11. DJF AR Contributions to poleward latent heat transport (LHT)**. AR contributions to poleward LHT in DJF for the test dataset years (1984, 1997, 2003, and 2010) in **(a)** MERRA2 and **(b)** ECMWF-IFS-HR. AR-induced LHT is shown at consensus levels of 0.1 (dashed green line) and 0.8 (dashed blue line). The best estimate (dashed yellow line) of AR-induced LHT is the poleward LHT multiplied by ARCI, effectively weighing LHT by AR detection confidence.

In Figure 11, we show that AR-induced poleward LHT varies widely for different levels of ARTMIP consensus. More permissive levels of ARCI (e.g. a value of 0.1, or 10% consensus) indicate that ARs account for virtually all the total LHT. At many latitudes, ARs even surpass the total LHT. While total LHT includes both a poleward and equatorward LHT, ARs are primarily associated with the poleward component; the magnitude of total LHT may thus be smaller than the magnitude of AR-induced LHT. With higher levels of ARTMIP consensus, such as 80 percent, ARs only play a small role in total meridional LHT

(Figures C1b and 11b). Because of the large variation in AR-induced poleward LHT, detector uncertainty plays an important role in ZN98's conclusion on the role of ARs in moisture flux. These results are in line with Newman et al. (2012). Using two AR detection algorithms available at the time, they (hereafter referred to as N12) note that AR size and spatial extent are sensitive to the choice of subjective parameters and thresholds in an AR detection algorithm. This sensitivity poses a major challenge for assessing meridional moisture transport induced by ARs.

As consensus increases, the spatial extent of ARs significantly decreases. (We visualize this for a sample time step in Figure 1, in which global coverage of ARs changed significantly as AR consensus increased.) This change in spatial extent causes the wide variation in estimates of AR-induced LHT. If the detected ARs account for a small portion of the globe, then their implied LHT will also be small. In Figures 11a and 11b, we calculate a "Best Estimate." We weigh the poleward LHT fields by the probabilistic AR detections. In effect, this gives an estimate of AR-induced LHT while taking detector uncertainty

into account. In this estimate, each detector is weighed evenly. Using the best estimates, ARs account for the majority of the poleward LHT near its peak in the midlatitudes. There is a hemispheric asymmetry, with ARs in the Southern Hemisphere



accounting for more of the poleward LHT than ARs in the Northern Hemisphere. This could also be due to the fact that there are more algorithms run in the Northern Hemisphere than in the Southern Hemisphere, since some of the algorithms in the ARCI dataset were only run on specific regions.

An important benefit of our training setup is the exploration of detection uncertainty. Using the ARCI dataset, as opposed to binary AR detection labels, enables this capability. The CNN-based estimates of AR-induced meridional LHT are consistent with those from the ARCI dataset. Figure 11a is very similar to the analogous quantities from ARTMIP (Figure C1). Since these quantities are calculated on the years in the ARCNN's test set, the similarity between Figure 11 and Figure C1 serves as further validation of the ARCNN's performance. The similarily between these figures indicates that the ARCNN is correctly replicating the spread of AR detections from different detectors in ARCI.

ARs' contributions to poleward LHT is largely consistent across MERRA2 and ECMWF-IFS-HR, as shown by the similarity of Figures 11a and 11b. In the midlatitudes, ECMWF-IFS-HR and MERRA2 have similar amounts of poleward LHT. Additionally, the ARCNN detections show a large variation in AR-induced total meridional LHT between consensus thresholds of 80% and 10%. One notable difference between ECMWF-IFS-HR and MERRA2 is that the former has larger poleward transport at the southern edge of the Hadley Cell, at approximately 22 degrees south (Figures 11a and c). This difference is related to how ECMWF-IFS-HR resolves the Hadley Cell and the Mean Meridional Circulation (not shown), so it is likely not related to ARs. Additionally, peak LHT near 40°N is about 25% smaller in ECMWF-IFS-HR than in MERRA2, with a similar difference at 45°S. This is another difference between the high-resolution climate model and the historical climate reanalysis; see Donohoe et al. (2020) for a detailed comparison of meridional heat transport between CMIP5 models (which are lower resolution than ECMWF-IFS-HR) and reanalysis. Understanding the reason for these differences is out of the scope of this study.

## 5   Discussion and Conclusions

ARs serve a crucial role in the climate system, and they have significant consequences for human systems. Here, we train multiple CNNs to detect ARs in different climate datasets (gridded satellite data, three reanalyses [MERRA2, ERA-Interim, and the ERA-20th Century Reanalysis], and output from two climate models [ECMWF's HighResMIP high-resolution IFS dataset, and a 0.25°-horizontal resolution CAM5]), different physical variables (IWV, IVT, and brightness temperature), and multiple horizontal resolutions (ranging from 0.25° to 1.25°). These ARCNNs' detections encode detector uncertainty because they are trained on the AR Consensus Index. Because they generalize to new datasets, they enable studying ARs in a variety of contexts. We compare detections from ARCNNs and ARCI using metrics such as IoU (Figure 8), the spatial patterns of AR frequency (Figure 7), and AR-induced latent heat transport (Figure 11 and Figure C1). To validate that the ARCNN is getting the right answer for the right reason, we use an idealized simulation. We design a shallow water simulation in which we expect virtually all the vapor transport to be attributed to AR-like features. The ARCNNs' detections are consistent with this expectation.





In this study, we consider uncertainty due to the choice of ARTMIP detector. We do not consider the uncertainty introduced
by the CNNs themselves. The CNN uncertainty can be caused by the choice of CNN architecture, optimization method, and
weight initialization (Gawlikowski et al., 2021; Gal and Ghahramani, 2015), among other factors. Future research is necessary
to explore this source of uncertainty. This would enable a detailed decomposition of the ARCI uncertainty and the uncertainty
from the CNNs themselves.

The ARCNNs presented here enable analysis of ARs in a variety of climate datasets, including those without associated
ARTMIP labeling campaigns. With style transfer, ARCNNs can scale to new datasets without requiring new AR labeling
campaigns. These labeling campaigns serve as a significant bottleneck, as they require expensive time from experts to hand-
draw ARs or re-tune ARTMIP algorithms for new datasets. In Figure 8, we show the performance of five different ARCNNs;
each ARCNN enables a specific form of generalization. For instance, one ARCNN detects ARs in IWV and IVT, a different
ARCNN detects ARs in MERRA2 and ERA-I, one detects ARs in ECMWF-IFS-HR and ERA-I, etc. Future work is necessary
to train one ARCNN to perform all forms of generalization: across input fields, resolutions, and datasets. Such an ARCNN
could be readily applied to large amounts of climate data, and its performance could be even more generalizable than the ones
presented in this manuscript. In particular, this could enable AR training and detection in the full multimodel ensemble of
HighResMIP. Training this ARCNN would require more computational power than was available in this study. We only used 1
GPU for training, and we had to use a small batch size (2 training samples per batch) due to GPU memory limitations. A larger
batch size would make it possible to iterate through the training data faster. With more GPUs available in high-performance
computing centers, it would be possible to train an ARCNN on large amounts of data and parallelize the training in a distributed
fashion (Kurth et al., 2018; Sergeev and Del Balso, 2018).

Here, our ARCNN generalizes to different *inputs*, but an important future avenue concerns training on different *labels*.
ClimateNet (Prabhat et al., 2021) include hand-drawn AR detections by experts, and O'Brien et al. (2020) assemble a dataset
of AR counts by experts. Future work is necessary to adapt the semi-supervised framework introduced here to ARCNNs trained
on those datasets. With future ARCNNs that replicate hand-identified ARs on large datasets, it would be possible to quantify
detector uncertainty in threshold-based AR detectors, hand-drawn AR detectors, and manual AR counts. These labeled datasets
are heterogeneous: ClimateNet includes hand-drawn AR contours on a historical simulation of C20C-CAM5, and the ARCI
dataset is available on MERRA2. There are additional threshold-based labels available on C20C-CAM5, CMIP5, and CMIP6
through additional ARTMIP experiments. The size of these datasets vary significantly: ClimateNet includes hand-drawn AR
contours on 300 time stamps, whereas the ARTMIP catalogues are several orders of magnitude larger. It would be a fruitful
machine learning exercise to combine these heterogeneous labels and data sources into training one ARCNN. In particular, this
effort could leverage advances in learning under noisy labels (Song et al., 2020).

In this manuscript, we use ARCNNs to explore the role of ARs in the climate system. While ZN98 find that ARs are re-
sponsible for virtually all the extratropical moisture flux, we find that AR detector uncertainty plays a significant role in this
conclusion. For a given level of detector consensus, there is significant variation in the contribution of ARs to poleward LHT
(Figure 11). While ARCI provides a "best estimate" of AR contributions to LHT, the underlying detection algorithms use corre-
lated thresholds in IVT or length scale; the algorithms are not independent. Therefore, a simple majority vote (51% consensus)





may not be the most appropriate estimate of AR-induced latent heat transport. Further investigation of the relationship between ARs, transient eddies, and meridional LHT is necessary. By understanding and characterizing the relationship between these phenomena, in future work, we plan to develop a physically grounded reference quantity for AR-induced LHT. With this quantity, it would be possible to constrain the suite of ARTMIP algorithms or develop a weighed ARTMIP ensemble. Additionally, this quantity could also be incorporated into the training of the CNN directly. The CNN could be trained to identify ARs based on the ARCI training dataset but with an additional constraint: its identified ARs must match a given reference quantity about AR-induced poleward energy transport. In order to train the CNNs in this way, it would be useful to leverage methods from physically informed machine learning in climate science (Kashinath et al., 2021).

Ultimately, these ARCNNs can be used to study how ARs will change in the future. Further research is necessary to use the ARCNN on future scenarios of climate models. The AR detections from an ARCNN trained with style transfer could be compared to AR detection algorithms used in the ARTMIP Tier2 Intercomparison, which uses existing AR detectors on CMIP5 and CMIP6 climate models. Through considerations of CNN uncertainty, AR detector uncertainty, and climate model uncertainty, future research is necessary to explore rigorously how ARs and their contributions to LHT will change in the future.

In machine learning for climate science, a major challenge is that neural networks must generalize to new climate datasets and scenarios (Reichstein et al., 2019; Beucler et al., 2021). Machine learning has been used to emulate processes such as atmospheric convection (Rasp et al., 2018; O'Gorman and Dwyer, 2018; Beucler et al., 2020) and radiative transfer (Cachay et al., 2021), to detect phenomena such as weather fronts (Dagon et al., 2022), thunderstorms (Molina et al., 2021), and clouds (Schulz et al., 2020), and to forecast climate responses to various forcings (Watson-Parris et al., 2022). Also, ML offers a promising avenue for assimilating heterogeneous modeled and observed datasets (Geer, 2021; Brajard et al., 2021; Gettelman et al., 2022), many of which are at a range of resolutions and complexities (Scher and Messori, 2019; Yuval and O'Gorman, 2020; Yuval et al., 2021). In all these research areas, machine learning needs to generalize to out-of-sample or out-of-distribution data. As ML is increasingly deployed in climate-related fields, it is crucial that ML models maintain accurate performance on a variety of climate scenarios or datasets. Style transfer, a form of domain adaptation, can be a useful tool in this generalization. **In Appendix Section B, we include a guide and link to a tutorial (with code) explaining how to apply the methods introduced here to other problems in climate and atmospheric science.**

Another priority in applying neural networks to climate datasets is trustworthy and explainable AI (XAI). Mamalakis et al. (2022b) have developed statistical benchmarks to evaluate different XAI methods. XAI has been used to study precursors to El Niño (Ham et al., 2019; Toms et al., 2020), the mechanisms of extreme precipitation (Davenport and Diffenbaugh, 2021), and subseasonal prediction (Mayer and Barnes, 2021). To explore these scientific research areas, a hierarchy of climate models can be used to evaluate CNN explainability. In atmospheric science, the rich hierarchy of climate models make them a ripe testing ground to evaluate CNNs.

We conclude that style transfer and a model hierarchy are two key tools in this study used to generalize and validate ARC-NNs. These two tools are broadly applicable to a variety of research in machine learning in climate science. In this manuscript, we use them to address the shortfalls of existing heuristic and CNN-based detectors. We present a rigorously validated, gener-



alizable AR detector that preserves uncertainty introduced by the ensemble of AR detection algorithms. Style transfer, climate
model hierarchies, and latent heat transport offer a promising path forward to reduce the spread in AR detections. This will
enable more confident estimates of future changes in AR frequency and intensity as the climate changes.

*Code and data availability.* We open-source nine datasets:

1. the ARCI labels on MERRA2 IWV: https://portal.nersc.gov/archive/home/a/amahesh/www/artmip_probabilistic_labels_all_vars.tar

2. the ARCI labels on MERRA2 IVT (same link as above)

3. the ARCI labels on GRIDSAT: https://portal.nersc.gov/archive/home/a/amahesh/www/era_20cr.tar: https://portal.nersc.gov/archive/home/a/amahesh/www/GRIDSAT.tar,

4. the ARCI labels on ERA-I IVT: https://portal.nersc.gov/archive/home/a/amahesh/www/era_probabilistic_labels.tar

5. the ARCI labels on ERA20th Century Reanalysis IVT: https://portal.nersc.gov/archive/home/a/amahesh/www/era_20cr.tar

6. ARCNN detections in MERRA2 IVT: https://portal.nersc.gov/archive/home/a/amahesh/www/merra2_nn_preds.tar. These predictions use the ARCNN Experiment 1 in Table 1.

7. ECMWF-IFS-HR IVT: https://portal.nersc.gov/archive/home/a/amahesh/www/HighResMIP-ECMWF-IFS-HR_IVT.tar

8. ARCNN detections in ECMWF-IFS-HR IVT: https://portal.nersc.gov/archive/home/a/amahesh/www/ecmwf-ifs-hr_nn_preds.tar. These predictions use the ARCNN Experiment 6 in Table 1.

9. The idealized single layer simulation: https://portal.nersc.gov/archive/home/a/amahesh/www/idealized_ar.tar

At the Zenodo DOI (https://doi.org/10.5281/zenodo.7814401), we also open-source code for three tasks: our code to run the *Atmospheric Rivers* textbook threshold algorithm, our code to run the idealized climate simulation, and **a tutorial of how to use our loss function and models**. We recommend the tutorial as a starting point to understand how to use the loss function to train your own CNN.

We also open-source the learned parameters for six trained ARCNNs: one for MERRA2 (using either IWV or IVT as input), one for GRIDSAT (using the brightness temperatures in the infrared window), one for ERA-I (using IVT as input), one for ERA20th Century Reanalysis (using IVT as input), one for C20C CAM5 (using IVT as input), and one for ECMWF-IFS-HR (using IVT as input). In the above tutorial, we include instructions on how to load the model (using Pytorch) and generate AR detections with it. These are available at the above Zenodo repository under the ar_segmentation_tutorial/trained_models folder.

*Video supplement.* We provide a visualization of the shallow water simulation on YouTube for convenience: https://youtube.com/watch?v=7Gq7e5PIRio. For archival purposes, the video is also stored at https://doi.org/10.5281/zenodo.7806480.



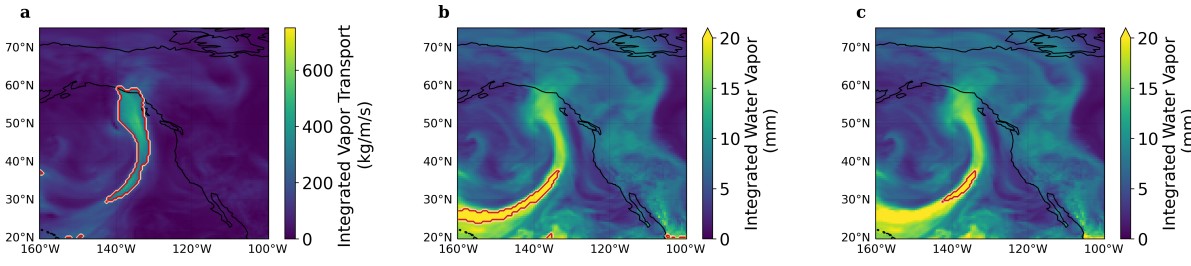

**Figure A1. Further info on the textbook algorithm**. In **(a)**, Integrated Vapor Transport at the same time as Figure 2. The dashed red contour shows the region above 250 kg/m/s. **(b)** shows Integrated Water Vapor. The dashed red contour denotes the region above 20 mm. **(c)** shows Integrated Water Vapor, and the dashed red line denotes the region both above 250 kg/m/s and 20 mm.

## Appendix A: Explanation of Threshold-based AR detector from AR Textbook

In the *Atmospheric Rivers* textbook (Ralph et al., 2020), ARs are defined as objects above 250 kg/m/s of IVT and 20 mm of IWV, with a length scale of at least 2000 km. We define the length scale as the distance along the diagonal of the bounding box of the identified AR. This definition is similar to that of Goldenson et al. (2018), who define the AR length as the distance between the northeast and southwest corner of an AR.

Figure A1 shows that the overlap between the regions above 250 kg/m/s and 20 mm (Fig A1c) is relatively small. Thus, the identified object does not meet the minimum length scale for an AR, and the algorithm misses the AR. The IWV threshold causes the algorithm to miss the part of the AR that makes landfall because that region is just under 20 mm of IWV. The IVT threshold causes the algorithm to miss the southern tip of the AR originating in lower latitudes.

There could be other ways of defining the AR length scale. For instance, Guan and Waliser (2015) calculate the length along the major axis of the AR. Because the region where both IWV > 20 mm and IVT > 250 kg/m/s (Fig A1c) is small, it is likely that other length scale definitions would also subset out this feature. The distance between the southwest and northeast corner of the AR is approximately 1600 km, well under the minimum length scale of 2000 km. Even if some length scale definitions would calculate this as an AR, the identified region misses crucial aspects of the AR, such as the section that makes landfall in southeast Alaska and northwest Canada.

## Appendix B: How To Apply Style Transfer to Other Problems in Climate and Atmospheric Science

In order to use the AR detectors trained here, the Toolkit for Extreme Climate Analysis (TECA: https://teca.readthedocs.io/en/latest/) can be deployed for AR detection in parallel. TECA includes support for GPU and CPU-based extreme climate event analysis on high-performance computing machines; it fully leverages parallel computing to perform AR detections in parallel across many timesteps of large datasets. Additionally, we open-source the learned weights of our ARCNNs, which we trained using Pytorch. Using Pytorch and these learned weights, it can be possible to readily run our trained AR detectors: the AR detectors do not have to be retrained from scratch.



Additionally, in this manuscript, we present a semi-supervised framework for learning across domains, including different models, datasets, input fields, and resolutions. In a semi-supervised framework, the CNN is learning from both labeled data and unlabeled data. To apply this framework to a new problem in climate and atmospheric science, we recommend the following process:

1. Identify the *labeled* dataset. Each sample in this dataset is composed of an input-label pair, and the neural network learns to approximate the function to transform the input into the label. Here, our labeled datasets were from MERRA2, ERA-I, GRIDSAT, and ERA20th Century.

2. Identify the *unlabeled* dataset. In this manuscript, our unlabeled dataset is the ECMWF-IFS-HR model. Through this semi-supervised framework, the goal is to train a CNN that performs well on the unlabeled dataset, even though no labels are available.

3. Change the loss function. Commonly, in supervised learning contexts, the loss function only minimizes the error between the CNN's predictions and the labels. In this semi-supervised learning framework, the loss function minimizes the *content loss* and *style loss* between the input and labels, and it minimizes an *additional style loss* between the predictions on the unlabeled dataset and the labels. This ensures that the Gram matrix of a compressed representation of the CNN's predictions are the same on the labeled dataset and unlabeled dataset. See Figure B1 for a diagram of these loss functions.

In order to apply this framework to new problems in atmospheric science, the primary change is to the loss function, as described in step 3 above. Along with the code to run our experiments, we also provide a sample Python tutorial under the `tutorials` folder in our code repository. **We include a Jupyter notebook tutorial of how to use the loss functions presented here.** With Python code, this tutorial explains how to implement semi-supervised style transfer. We intend this tutorial to be of use in applying these methods to other research problems in climate and atmospheric science.

## Appendix C: Effect of ARTMIP uncertainty on AR-Induced Latent Heat Transport

In Figure C1, we use the ARCI dataset to show how AR detector uncertainty affects estimates of AR-induced latent heat transport. The "Best Estimate," 10% Consenus Threshold, and 80% Conensus Threshold lines in Figure C1 are very similar to those of the ARCNN (Figure 11). Since both figures cover the years in the ARCNN's test dataset, this serves as further validation of the ARCNN. Its predictions closely match those from ARCI.

## Appendix D: ARTMIP algorithms

For the ARCI dataset on MERRA2, ERA-I, ERA20th Century Reanalysis, and GRIDSAT, we use the following algorithms to assemble the ARCI dataset from ARTMIP tier1 (Christine Shields, Shields@Ucar.Edu, 2018): Gershunov (Gershunov et al., 2017), Lora_Global, Lora_Npac (Lora et al., 2017), Mundhenk, Rutz (Rutz et al., 2014), PNNL1_Hagos (Hagos et al., 2015), PNNL2_lq (Leung and Qian, 2009), Goldenson (Goldenson et al., 2018), Mundhenk (Mundhenk et al., 2016), Payne (Payne



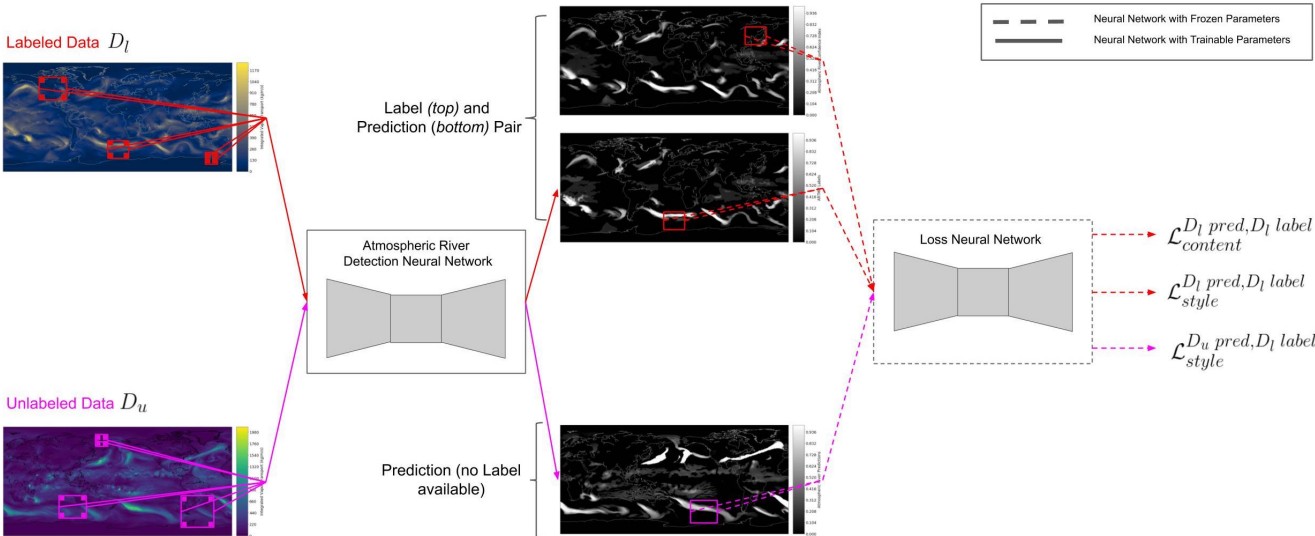

**Figure B1.** Training Scheme Diagram. This diagram illustrates the training scheme used to teach a neural network to identify atmospheric rivers in labeled climate datasets ($D_l$) and unlabeled climate datasets ($D_u$). A climate dataset is labeled if there are pre-identified ARs (such as the ARCI) available. On the labeled climate datasets, the AR Detection Neural Network minimizes the content loss between its predictions and the labels and the style loss between its predictions and the labels. On unlabeled climate datasets, the AR Detection Neural Network minimizes the style loss between its predictions and labels from the labeled dataset.

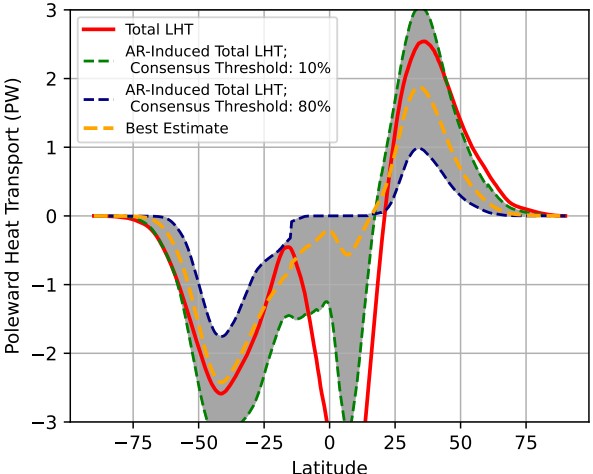

**Figure C1.** Same as top row of Figure 11, but using the ARCI instead of ARCNN detections.



and Magnusdottir, 2015), CONNECT700 (Sellars et al., 2015), CONNECT500, Walton, GuanWaliser (Guan and Waliser, 2015), and tempest (Ullrich and Zarzycki, 2017; McClenny et al., 2020),

Using style transfer, we train an ARCNN to detect ARs in CAM5. This ARCNN was not trained directly on ARTMIP labels in CAM5; rather, it used the semi-supervised learning framework described in Section 2.4. After the model was trained, we compared the ARCNN's detections with ARTMIP algorithms: Gershunov (Gershunov et al., 2017), Lorav2 (Lora et al., 2017), Goldenson (Goldenson et al., 2018), Payne (Payne and Magnusdottir, 2015), tempest_IVT250 (Ullrich and Zarzycki, 2017; McClenny et al., 2020), tempest_IVT500, and tempest_IVT700. We calculated IoU scores using the above datasets.

*Author contributions.* AM, TAO, WB, BL, AE, and WDC performed research and wrote the manuscript. BL, AM, AE, and TAO integrated the CNNs into the Toolkit for Extreme Climate Analysis, an open-source software package for analyzing climate datasets using high-performance computing. Documentation for TECA can be found at https://teca.readthedocs.io/en/latest/, and the method to detect ARs using the DeepLabv3+ CNN trained here is called *teca_deeplab_ar_detect*.

*Competing interests.* The authors declare they have no competing interests.

*Acknowledgements.* ARTMIP is a grassroots community effort that includes a collection of international researchers from universities, laboratories, and agencies. Co-chairs and committee members include Jonathan Rutz, Christine Shields, L. Ruby Leung, F. Martin Ralph, Michael Wehner, Ashley Payne, Travis O'Brien, and Allison Collow. Details on catalogs developers can be found on the ARTMIP website (http://www.cgd.ucar.edu/projects/artmip). ARTMIP has received support from the US Department of Energy Office of Science Biological and Environmental Research (BER) as part of the Regional and Global Climate Modeling program, and the Center for Western Weather and
Water Extremes (CW3E) at Scripps Institute for Oceanography at the University of California, San Diego.



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
