# Peer review of "Identifying Atmospheric Rivers and their Poleward Latent Heat Transport with Generalizable Neural Networks: ARCNNv1"

_EGUsphere, 2023_

## Referee Comment (RC1)

**Title**: Identifying Atmospheric Rivers and their Poleward Latent Heat Transport with Generalizable Neural Networks: ARCNNv1

Authors: Ankur Mahesh, Travis A. O'Brien, Burlen Loring, Abdelrahman Elbashandy, William Boos, William D. Collins

**Summary:**

This paper is divided into two parts. First, Mahesh et al. describe a new machine learning methodology to identify atmospheric rivers (ARs) based on convolutional neural networks which utilizes a semi-supervised framework and image-style transfer learning. Uncertainty is quantified as well as the performance of the new methodology using a multi-pronged approach which includes observations, simplified models, and complex models. Second, latent heat transport attributable to ARs is quantified and demonstrated to have large uncertainty due to detection technique.

**Overall Comments:**

Mahesh et al. demonstrate a deep understanding of the problems and issues surrounding AR detection and the subsequent consequences on characterization of a physical process, which in this case, focus on latent heat transport. The paper provides thorough and robust quantification of uncertainty by leveraging many different datasets such as the ARTMIP database, reanalysis products, satellite data, and both climate and idealized simulations. Additionally, they provide details on computational resources, code, and datasets, all of which is necessary for reproducibility and is appreciated. I find the results presented here quite convincing and robustly vetted. Their conclusions on the spread of uncertainty due to detectors for latent heat transport is an important contribution and nicely updates and improves upon the current state of the literature on the topic. I recommend publication after a few minor comments and questions are answered. It was a pleasure to read and review.

**Specific Comments:**

Line 21: I think this statement has been demonstrated by all the main ARTMIP papers (Rutz et al, 2019, Collow et al., 2021, Shields, Payne et al., 2023).

Line 32: Shields, Payne et al. 2023 should also be added to the list for climate change ARDT comparisons.

Shields, C. A., Payne, A. E., Shearer, E. J., Wehner, M. F., O'Brien, T. A., Rutz, J. J., Leung,
L.R., Ralph, F. M., Collow, A. B. M., Ullrich, P. A. Ullrich, Dong, Q., Gershunov, A.,
Griffith, H., Guan, B., Lora, J. M., Lu, M., McClenny, E., Nardi, K. M., Pan, M., Qian, Y.,
Ramos, A. M. Ramos, Shulgina, T., Viale, M., Sarangi, C., Tomé, R., Zarzycki, C. (2023).
Future atmospheric rivers and impacts on precipitation: Overview of the ARTMIP Tier 2
high-resolution global warming experiment. Geophysical Research Letters, 50, e2022GL102091.
https://doi.org/10.1029/2022GL102091

Line 95: Are there biases in the models that would impact AR detection? IWV, IVT? If so, a sentence or two addressing these would be helpful. Why not ERA5?

Line 105: Rutz et al. 2019 should be included in the ARTMIP list.

Figure 3: Confidence index (plot y label), or Consensus index (Figure caption)?

Section 2.1: I really like the ARCI and think its application in your paper is appropriate given you are looking at heat transport via mid-latitude ARs. However, one limitation I see is for regions such as the poles, where the majority of ARDTs actually don't capture ARs reaching either into the Arctic, or on the Antarctic continent correctly compared to ARDTs designed for high latitudes (Shields et al., 2022). The ARCI might not be that useful here because many of the globals (with no polar constraints) are not "fit for purpose". I'd recommend a qualifying statement on the use of ARCI for middle latitudes versus polar regions.

Shields, C. A., Wille, J. D., Marquardt Collow, A. B., Maclennan, M., & Gorodetskaya, I. V. (2022). Evaluating uncertainty and modes of variability for Antarctic atmospheric rivers. Geophysical Research Letters, 49, e2022GL099577 . https://doi.org/10.1029/2022GL099577.

Line 134: I know you include this in the discussion, but I think a line about how this is different from ClimateNet is needed here as well. My guess is there will be readers that are undoubtedly familiar with ClimateNet, given your use of the same underlying CAM5 data and DeepLabv3+ code? ( i.e., your use of ARCI vs ClimateNet's hand drawn labels, or perhaps I am misunderstanding something)?

Line 280: For my clarification: Is the reverse also true? I.e., if the neural network is trained on *model* data, then applied to *reanalysis*, the same problem would exist? Isn't this what ClimateNet's ARTMIP contribution does? Maybe you don't have access to that answer, but if this is the case, would the ClimateNet ARTMIP catalogues have these same problems? Is this concerning that they are included in your ARCI?

Lines 451-454: For 11b, the 80% consensus line is actually bigger in the NH. Maybe move your explanation of this in lines 465-471 after this initial hemispheric asymmetry statement.

Figure 11: I really like this figure! Why not add Figure C1 as Figure 11c? It is a nice demonstration of the validation of your ARCNN. Have you looked at other energy transport quantities like sensible heat?

Figure C1 label: Do you mean Figure 11a, rather than top row?

---

## Referee Comment (RC2)

Title: Identifying Atmospheric Rivers and their Poleward Latent Heat Transport with Generalizable Neural Networks: ARCNNv1

Authors: Ankur Mahesh, Travis A. O'Brien, Burlen Loring, Abdelrahman Elbashandy, William Boos, William D. Collins

**Summary:**

This work introduces various different CNN-based AR tracking methods with the goal of addressing the issue of AR detection uncertainty and flexibility of AR detection with different datasets and resolutions. An index composed of ARs detected by various other common methods is defined to validate the approach. A semi-supervised learning approach based on image-style transfer is applied. The result shows robust consistency with other common AR detection methods. The results also demonstrate that the ARCNNs also have consistent amounts of AR-induced latent heat transport with other common detection methods.

**Overall Comments:**

Mahesh et al. bring forward a highly effective method of addressing several existing challenges in AR-related research. The writing is clear and easy to follow. The authors provide easily accessible code and data to reproduce the results and apply them to other future studies. The results show high performance of the method. My largest concern about this work is the choice of ARTMIP methods used for validation. Out of seven methods used for validation (when calculating IoU), three of them are taken from the same group (Tempest). Choosing nearly half of detection methods in the validation set that are almost identical could cause the results to be misleading. There were also repeated ARTMIP methods used in the ARCI. Once that and several other comments are addressed, I recommend publication.

**Specific Comments:**

Line 66: I'm not convinced that different datasets would require new training labels for the purpose of detecting ARs. Re-gridding the training data could allow the user to have some flexibility with other datasets.

Line 134: You could justify the claim of strong performance with Wu et al. 2019

Wu, T., Tang, S., Zhang, R., & Zhang, Y. (2019). CGNet: A Light-weight Context Guided Network for Semantic Segmentation. ArXiv:1811.08201 [Cs]. Retrieved from http://arxiv.org/abs/1811.08201

Line 299: The language here ("its detected AR probabilities are too low") could be improved. Instead, I would suggest changing this to something along the lines of "its detected AR probabilities are lower than the ARCI".

Line 372-373: "CNNs have millions of tunable parameters" It could be useful to the reader to include a source for this claim

Line 631-632: Mundhenk is mentioned twice here. The first mention of Mundhenk does not include a reference so it is unclear if Mundhenk is being used twice, if there are two different versions used, or if this was a typo.

The ARDTs used for the AR Consensus Index include multiple algorithms from the same group (Lora, Mundhenk, CONNECT). While there are slight variations between different algorithms created by the same groups, some justification of the choice to weight algorithms from some groups more heavily than others in the ARCI could be useful.

Three different versions of Tempest are used to calculate IoU. Some justification for this could be useful.

It is unclear which version of Tempest is used in the ARCI

In Figure 8, it is unclear if the calculated IoU scores only representing grid points in which ARs are detected or is the background class IoU factored into the calculation as well.

I suggest referencing Higgins et al. 2023 to establish some precedent to using a variety of different ARTMIP labels to validate ARCNNs

Higgins, T. B., Subramanian, A. C., Graubner, A., Kapp-Schwoerer, L., Watson, P. A. G., Sparrow, S., et al. (2023). Using Deep Learning for an Analysis of Atmospheric Rivers in a High-Resolution Large Ensemble Climate Data Set. *Journal of Advances in Modeling Earth Systems*, 15(4), e2022MS003495. https://doi.org/10.1029/2022MS003495

---

## Author Response (AR1)

**Response to Reviewers: "Identifying Atmospheric Rivers and their Poleward Latent Heat Transport with Generalizable Neural Networks: ARCNNv1"**

November 30, 2023

**Overview**

We sincerely thank the reviewers for their constructive comments and review of our paper. These comments have substantively improved our manuscript. We have included responses to the reviewers' comments below, with the reviewer comments in black text and our response in green text. In addition to this document, we also will submit a revised version of the manuscript.

**Comments from Reviewer #1**

Mahesh et al. demonstrate a deep understanding of the problems and issues surrounding AR detection and the subsequent consequences on characterization of a physical process, which in this case, focus on latent heat transport. The paper provides thorough and robust quantification of uncertainty by leveraging many different datasets such as the ARTMIP database, reanalysis products, satellite data, and both climate and idealized simulations. Additionally, they provide details on computational resources, code, and datasets, all of which is necessary for reproducibility and is appreciated. I find the results presented here quite convincing and robustly vetted. Their conclusions on the spread of uncertainty due to detectors for latent heat transport is an important contribution and nicely updates and improves upon the current state of the literature on the topic. I recommend publication after a few minor comments and questions are answered. It was a pleasure to read and review.

Thank you very much for your review of our paper.

Line 21: I think this statement has been demonstrated by all the main ARTMIP papers (Rutz et al, 2019, Collow et al., 2021, Shields, Payne et al., 2023).

Thank you for the suggestion. We have amended the citations accordingly to cite the papers you suggest.

Line 32: Shields, Payne et al. 2023 should also be added to the list for climate change ARDT comparisons.

Thank you very much for this pointer. We have added the citation.

Line 105: Rutz et al. 2019 should be included in the ARTMIP list.

We have added this paper to the ARTMIP list.

Figure 3: Confidence index (plot y label), or Consensus index (Figure caption)?

Thank you for catching this error. The AR Consensus Index is the correct term, and we have changed the label of the figure accordingly.

Section 2.1: I really like the ARCI and think its application in your paper is appropriate given you are looking at heat transport via mid-latitude ARs. However, one limitation I see is for regions such as the poles, where the majority of ARDTs actually don't capture ARs reaching either into the Arctic, or on the Antarctic continent correctly compared to ARDTs designed for high latitudes (Shields et al., 2022). The ARCI might not be that useful here because many of the globals (with no polar constraints) are not "fit for purpose". I'd recommend a qualifying statement on the use of ARCI for middle latitudes versus polar regions.

This is an excellent point. We have included a citation to the recommended paper, and we have added a sentence at the end of section 2.1 explaining the use of the ARCI, given our focus on midlatitude heat transport. This starts at line 129 of the updated manuscript:

*Shields et. al. (2022) demonstrate that global ARTMIP algorithms may not correctly identify ARs in polar regions, such as the ice sheets in East Antarctica; they note that Antarctic-specific AR detection tools are necessary for these regions. Therefore, in this manuscript, we focus on midlatitude ARs and their associated heat transport.*

Line 134: I know you include this in the discussion, but I think a line about how this is different from ClimateNet is needed here as well. My guess is there will be readers that are undoubtedly familiar with ClimateNet, given your use of the same underlying CAM5 data and DeepLabv3+ code? ( i.e., your use of ARCI vs ClimateNet's hand drawn labels, or perhaps I am misunderstanding something)?

We have included a sentence contrasting our work to ClimateNet. It is correct that our work uses ARCI, compared to ClimateNet's use of hand-drawn labels. ARCI is probabilistic and is based on ARTMIP tier1 labels (originally run on MERRA2), whereas ClimateNet makes binary detections and uses CAM5 as its underlying input dataset. Our work also includes explicit changes to the loss function to generalize neural networks to different datasets, and we validate the neural network on an idealized experiment, where the ARs can be unambiguously determined.

Line 143-145 of the revised manuscript: *ClimateNet also uses successfully uses the DeepLabv3+ architecture for AR detection in CAM5. Here, we extend the use of DeepLabv3+ for probabilistic, rather than binary, AR detection with ARCI in MERRA2.*

Figure 11: I really like this figure! Why not add Figure C1 as Figure 11c? It is a nice demonstration of the validation of your ARCNN.

Thank you! This is a great suggestion. We have done so. We have moved Figure C1 to Figure 11 and changed the label and main text accordingly.

Line 280: For my clarification: Is the reverse also true? I.e., if the neural network is trained on model data, then applied to reanalysis, the same problem would exist? Isn't this what ClimateNet's ARTMIP contribution does? Maybe you don't have access to that answer, but if this is the case, would the ClimateNet

ARTMIP catalogues have these same problems? Is this concerning that they are included in your ARCI?

We do not include the ClimateNet catalogue in the ARCI, since we chose to avoid training an ARCNN on the output of another neural network. We are unsure if the same problems would exist if ClimateNet is applied to different datasets. This is because ClimateNet uses different AR labels (hand-drawn AR labels, as opposed to ARCI) and is evaluated for classification, as opposed to probabilistic AR detections. The hand-drawn AR labels is significantly smaller than the ARCI dataset. In some versions, ClimateNet also uses a different underlying neural network architecture (CGNET) Kapp-Schwoerer et al. [2020] and loss function (Jaccard loss). Given the different learning setups, we cannot immediately compare the generalizability between ClimateNet and the ARCNNs. In at least one instance, we note that ClimateNet's detected AR frequencies vary between two datasets (ERA5 and MERRA2): see Figure 3 of Collow et al. [2022]. This difference between ERA5 and MERRA2 surpasses that of many other ARTMIP algorithms. Broadly, distribution shift and domain generalization are very active areas in machine learning research Wang et al. [2022]. As the use of machine learning grows in climate change science, we anticipate that the challenge of generalization will arise, as it has in other fields, such as computer vision. We hope that the methods presented here can be applied to a variety of climate-related research areas.

Lines 451-454: For 11b, the 80% consensus line is actually bigger in the NH. Maybe move your explanation of this in lines 465-471 after this initial hemispheric asymmetry statement.

We removed the lines below from the revised manuscript. We think the hemispheric asymmetry in ARs, AR detector uncertainty, and AR latent heat transport is a deep topic, and we think this would be best explored in further research.

Now deleted: *There is a hemispheric asymmetry, with ARs in the Southern Hemisphere accounting for more of the poleward LHT than ARs in the Northern Hemisphere. This could also be due to the fact that there are more algorithms run in the Northern Hemisphere than in the Southern Hemisphere, since some of the algorithms in the ARCI dataset were only run on specific regions.*

Have you looked at other energy transport quantities like sensible heat?

This is a great suggestion. In this manuscript, we have not looked at AR-induced sensible heat transport or dry static energy transport. We agree that these are important topics to study in future research. We have added a sentence in the discussion idenitfying this as a topic to study for future work. In this manuscript, we focus on latent heat transport because of the role of ARs in the hydrological cycle and because of Zhu and Newell's initial statements regarding the role of ARs in extratropical moisture flux.

Line 542: *Additionally, future research is necessary to consider the role of ARs in sensible heat transport in present and future climates.*

Figure C1 label: Do you mean Figure 11a, rather than top row?

Thank you, this is absolutely correct. We have made the appropriate change. We note that we have included this figure with Figure 11 now, in line with the recommendation above.

**Comments from Reviewer #2**

Mahesh et al. bring forward a highly effective method of addressing several existing challenges in AR-related research. The writing is clear and easy to follow. The authors provide easily accessible code and

data to reproduce the results and apply them to other future studies. The results show high performance of the method.

Thank you very much for your review and for providing these overall comments.

My largest concern about this work is the choice of ARTMIP methods used for validation. Out of seven methods used for validation (when calculating IoU), three of them are taken from the same group (Tempest). Choosing nearly half of detection methods in the validation set that are almost identical could cause the results to be misleading.
Three different versions of Tempest are used to calculate IoU. Some justification for this could be useful.

Thank you for bringing up this point. To clarify, on MERRA2, GRIDSAT, ERA-I, and ERA 20th Century Reanalysis, we validate the ARCNN on **all** 14 algorithms in the ARCI: Gershunov, Lora_Global, Lora_Npac, Rutz, PNNL1_Hagos, PNNL2_lq, Goldenson, Mundhenk, Payne, CONNECT700, CONNECT500, Walton, GuanWaliser, and tempest. The IoU scores reported in Figure 8 on these input datasets is based on the ARCI from all these ARTMIP algorithms.

Not all ARTMIP algorithms that were run on MERRA2 were also run on CAM5. Therefore, only on CAM5, to develop a validation dataset, we used output from the 7 algorithms that were available during the time of the study: Gershunov, Lorav2, Goldenson, Payne, tempest_IVT250, tempest_IVT500, and tempest_IVT700. We have clarified the distinction between CAM5's validation dataset and MERRA2's validation in Appendix D.

Line 640: *To validate the performance on MERRA2, ERA-I, ERA20th Century Reanalysis, and GRIDSAT, we use these ARTMIP algorithms in our IoU score calculation.*

Line 646: *Not all algorithms used for the ARCI on MERRA2 were run on CAM5. Therefore, to validate the performance on the CAM5 dataset, we calculated the IoU between the prediction and the truth using these available datasets.*

In Peer Review Figure 1 of this document, we compare the result of the tempest algorithms using 250, 500, and 700 kg/m/s as their IVT threshold. At this time step, tempest250, tempest500, and tempest700 indicate that ARs cover 4.2% , 2.3% , and 0.8% of the globe, respectively. These three algorithms result in very different estimates of global AR activity. Therefore, we believe that these three algorithms are sufficiently different, so we use all three in our validation dataset.

Line 631-632: Mundhenk is mentioned twice here. The first mention of Mundhenk does not include a reference so it is unclear if Mundhenk is being used twice, if there are two different versions used, or if this was a typo.

Thank you for catching this. This was a typo. We used only one algorithm from Mundhenk in the ARCI. We erroneously listed it twice in the algorithm list, and we have correct this in the revised manuscript. We apologize for the confusion.

There were also repeated ARTMIP methods used in the ARCI. The ARDTs used for the AR Consensus Index include multiple algorithms from the same group (Lora, Mundhenk, CONNECT). While there are slight variations between different algorithms created by the same groups, some justification of the choice to weight algorithms from some groups more heavily than others in the ARCI could be useful.

[Figure]

Peer Review Figure 1: **Comparison of tempest250, tempest500, and tempest700 in CAM5**. (top left) IVT at a sample time step in the CAM5 simulation. AR detections from tempest250 (top right), tempest500 (bottom left), and tempest700 (bottom right).

Thank you for raising this issue, as it is a very important consideration related to AR detector uncertainty. Regarding CONNECT, we use two algorithms: connect500 and connect700. We choose to include both of these algorithms because we find that they have substantively different AR detections. In Peer Review Figure 2, we show that connect500 and connect700 yield different estimates of AR-induced LHT. At the latitude of peak AR-induced latent heat transport (LHT), connect500 identifies 1 PW more LHT in the Southern Hemisphere and 0.5 PW more LHT in the Northern Hemisphere than connect700. These are significant differences, considering the total LHT peaks around 2.5 PW. In Peer Review Figure 3, we highlight that connect500 detects two ARs that connect700 does not: one in the Atlantic Ocean in the Northern Hemisphere midlatitudes and one off the coast of South America. Connect700 does not identify these two ARs. At this time step, connect500 identifies 2.8% of the globe's area as having an AR, whereas connect700 identifies 0.8% of the globe's area to have an AR. (For reference, Peer Review Figure 3 in this document can be compared with Figures 1, 3, 4, 5, and 6 of the manuscript, as they all show the same time step.)

Reid et al. [2020] discuss the influence of different IVT and IWV thresholds on AR detection in depth, especially in Figure 4 and 6 of their paper. We also consider the effect of the interaction between an IVT threshold and other aspects of an AR detection algorithm (e.g. the shape requirement) in Figure 2 and Figure A1.

Regarding Mundhenk, we only use one algorithm from mundhenk (see comment above).

Regarding Lora, we use two algorithms from Lora: Lora_global and Lora_npac. The former is run for the whole globe, while the latter is only run in the North Pacific. Because of the different regional extent and considerations of these algorithms, we include both of them in the ARCI.

Line 66: I'm not convinced that different datasets would require new training labels for the purpose of detecting ARs. Re-gridding the training data could allow the user to have some flexibility with other

[Figure]

Peer Review Figure 2: **Comparison of AR-Induced Latent Heat Transport as indicated by connect500 and connect700**. Total LHT and AR-Induced LHT calculated for DJF of 1984.

165 datasets.

166 This is absolutely correct. In fact, we use this method to generate a training dataset of AR labels for
167 GRIDSAT (Line 66 of the original manuscript). Regridding the training data would not be possible for
168 detecting ARs in free-running climate simulations. This is because the individual time steps in a free-running
169 climate simulation do not align with those from observations or with each other. For this application, we
170 present style transfer in Line 66 of the original manuscript. This enables AR detection in ECMWF-IFS-HR.

171 Line 134: You could justify the claim of strong performance with Wu et al. 2019

172 We have cited Wu et. al. accordingly.

173 Line 299: The language here ("its detected AR probabilities are too low") could be improved. Instead,
174 I would suggest changing this to something along the lines of "its detected AR probabilities are lower than
175 the ARCI"

176 We have changed the text following this suggestion.

177 Line 307-308 of the revised manuscript: *its detected AR probabilities are consistently lower than those from*
178 *ARCI.*

[Figure]

[Figure]

[Figure]

[Figure]

Peer Review Figure 3: **Comparison of AR-Induced Latent Heat Transport as indicated by connect500 and connect700**. AR detections from 2009-04-01T00:00:00 are shown from Connect500 (left) and connect700(middle). Integrated Vapor Transport from MERRA2 (right) at this time step are shown.

Line 372-373: "CNNs have millions of tunable parameters" It could be useful to the reader to include a source for this claim.

We rephrased the statement to "millions of learned weights" for clarity. We have added Wu et. al. as a source. Figure 1 of Wu et. al. shows the number of weights that many architectures have.

It is unclear which version of Tempest is used in the ARCI.

We use the ARTMIP catalogue that has the identifier called "tempest." We have added a sentence here for clarification.

Line 649-650 of the revised manuscript: *The ARTMIP catalogues are organized by an ARTMIP algorithm identifier. The identifier of the algorithm used is written in quotations above.*

In Figure 8, it is unclear if the calculated IoU scores only representing grid points in which ARs are detected or is the background class IoU factored into the calculation as well.

We have made the appropriate clarification in the label of Figure 8. We calculate the IoU score in this way to represent false positives and false negatives in our metric.

Figure 8 caption: *The IoU scores are the average of the IoU for the foreground class (AR) and background class (not-AR).*

I suggest referencing Higgins et al. 2023 to establish some precedent to using a variety of different ARTMIP labels to validate ARCNNs.

Thank you very much for this pointer. We have included a reference to Higgins et. al. to the amended manuscript.

*Line 64: Higgins et. al. validate their neural network on ARTMIP algorithms, and they note that its performance is best when training and inference are performed on the same data domain and resolution.*

**References**

A. B. M. Collow, C. A. Shields, B. Guan, S. Kim, J. M. Lora, E. E. McClenny, K. Nardi, A. Payne, K. Reid, E. J. Shearer, R. Tomé, J. D. Wille, A. M. Ramos, I. V. Gorodetskaya, L. R. Leung, T. A. O'Brien, F. M.

Ralph, J. Rutz, P. A. Ullrich, and M. Wehner. An overview of artmip's tier 2 reanalysis intercomparison: Uncertainty in the detection of atmospheric rivers and their associated precipitation. *Journal of Geophysical Research: Atmospheres*, 127(8), Apr. 2022. ISSN 2169-8996. doi:10.1029/2021jd036155. URL http://dx.doi.org/10.1029/2021JD036155. 3

L. Kapp-Schwoerer, A. Graubner, S. Kim, and K. Kashinath. Spatio-temporal segmentation and tracking of weather patterns with light-weight neural networks. 2020. 3

K. J. Reid, A. D. King, T. P. Lane, and E. Short. The sensitivity of atmospheric river identification to integrated water vapor transport threshold, resolution, and regridding method. *Journal of Geophysical Research: Atmospheres*, 125(20), Oct. 2020. ISSN 2169-8996. doi:10.1029/2020jd032897. URL http://dx.doi.org/10.1029/2020JD032897. 5

J. Wang, C. Lan, C. Liu, Y. Ouyang, T. Qin, W. Lu, Y. Chen, W. Zeng, and P. Yu. Generalizing to unseen domains: A survey on domain generalization. *IEEE Transactions on Knowledge and Data Engineering*, 2022. 3

---

## Referee Report (RR1)

Reviewer #2 Response

Dear Mahesh et al.,

I am satisfied with the changes that you made since my last response. Congratulations on doing great work! I will recommend that the manuscript is published in its current form.

Best regards,
Reviewer #2